# T-POP: Test-Time Personalization with Online Preference Feedback

Zikun Qu [1 2]   Min Zhang [3]   Mingze Kong [1]   Xiang Li [4]   Zhiwei Shang [1]   Zhiyong Wang [5]   Yikun Ban [6]
Shuang Qiu [7]   Yao Shu [8]   Zhongxiang Dai [1]

## Abstract

Personalizing large language models (LLMs) to individual user preferences is a critical step beyond generating generically helpful responses. However, current personalization methods are ill-suited for new users, as they typically require either slow, resource-intensive fine-tuning or a substantial amount of pre-existing user data, creating a significant cold-start problem. To address this challenge, we introduce a new paradigm for real-time personalization by learning from online pairwise preference feedback collected during text generation. We propose T-POP (*Test-Time Personalization with Online Preference Feedback*), a novel algorithm that synergistically combines test-time alignment with *dueling bandits*. Without updating the LLM parameters, T-POP steers the decoding process of a frozen LLM by learning a reward function online that captures user preferences. By leveraging dueling bandits, T-POP intelligently queries the user to efficiently balance between exploring their preferences and exploiting the learned knowledge to generate personalized text. Extensive experiments demonstrate that T-POP achieves rapid and data-efficient personalization, significantly outperforming existing baselines and showing consistent improvement with more user interactions. Our code is publicly available at https://github.com/QuZikun/T-POP.

[1]The Chinese University of Hong Kong, Shenzhen [2]Shenzhen Loop Area Institute [3]East China Normal University [4]Tianjin University [5]The Chinese University of Hong Kong [6]Beihang University [7]City University of Hong Kong [8]The Hong Kong University of Science and Technology (Guangzhou). Correspondence to: Min Zhang <mzhang@cs.ecnu.edu.cn>, Zhongxiang Dai <daizhongxiang@cuhk.edu.cn>.

*Proceedings of the 43rd International Conference on Machine Learning*, Seoul, South Korea. PMLR 306, 2026. Copyright 2026 by the author(s).

## 1. Introduction

While large language models (LLMs) have achieved remarkable success in generating human-like text, a critical frontier remains: moving from generic, one-size-fits-all responses to deeply personalized interactions. Users increasingly expect models to understand and adapt to their unique voice, style, and preferences (Li et al., 2024a; Zhang et al., 2024; Li et al., 2025; Zhang et al., 2025b). The standard approach for aligning LLMs with human preferences has been through methods such as reinforcement learning from human feedback (RLHF) (Ouyang et al., 2022) and direct preference optimization (DPO) (Rafailov et al., 2023). However, these methods are primarily designed to align LLMs with *generic* human preferences, failing to capture the specific nuances of individual users.

To address this gap, some recent works have adapted the RLHF framework to align LLMs with the preferences of individual users (Jang et al., 2023; Li et al., 2024b; Park et al., 2024; Lee et al., 2024). While effective, these approaches necessitate fine-tuning the LLM parameters for each user. Consequently, they are often unable to adapt quickly and efficiently to new users, posing a significant barrier to scalability and real-time personalization.

In response to the limitations of fine-tuning, another line of research has focused on personalization methods that do not require parameter updates. These techniques include retrieval-augmented generation (RAG) to fetch user-specific information (Sun et al., 2024; Mysore et al., 2023; Salemi et al., 2024) and the integration of the historical data of the user directly into the LLM prompt (Kang et al., 2023; Liu et al., 2023; Li et al., 2024a; Kim & Yang, 2024). A common prerequisite for these methods, however, is the availability of sufficient user data. This leaves them inapplicable to new users for whom such data has not yet been collected, a critical challenge in the field of personalization known as the *cold-start* problem (Zhang et al., 2024).

To resolve this problem, a natural solution is to *collect user data online* for new users. Drawing from the widespread success of RLHF and DPO, the most reliable and easily provided form of user data is *preference feedback*, where users indicate their relative preference between a pair of LLM-generated responses. We therefore propose to collect

pairwise user preference data online to facilitate rapid personalization. This approach, however, introduces a crucial challenge: *how do we simultaneously (1) collect user preference data online and (2) use these sequentially available data to achieve effective personalization?*

In this work, we tackle this challenge by proposing a principled combination of *test-time alignment* (Khanov et al., 2024) and *dueling bandits* (Verma et al., 2024). We introduce our Test-Time Personalization with Online Preference *Feedback* (T−POP) algorithm, which is illustrated in Fig. 1. Following the test-time alignment paradigm, T−POP adjusts the decoding process of a frozen LLM via an additive reward function that captures user personalization. This reward function is learned online and assigns higher values to responses that are better aligned with the personal preferences of the user. To learn this reward function effectively, we incorporate dueling bandits into the token selection process, which allows us to strategically select a pair of candidate tokens at every decoding step to query the user for feedback. Thanks to the inherent ability of dueling bandit algorithms to balance exploration and exploitation, our T−POP is able to simultaneously (1) generate high-reward responses that are increasingly aligned with user preference (i.e., exploitation) and (2) collect diverse preference data to rapidly refine the reward function (i.e., exploration). As a result, T−POP achieves effective user personalization using only a small number of online user feedback interactions.

In summary, our main contributions are:

- We formalize the problem of test-time personalization with online preference feedback, addressing the critical cold-start challenge for new users.
- We propose T−POP, a novel algorithm that synergistically combines test-time alignment with dueling bandits to achieve rapid, data-efficient personalization without any parameter fine-tuning.
- Experiments show that T−POP significantly outperforms existing personalization baselines, with its effectiveness steadily increasing as more user feedback is provided.

## 2. Preliminary

**Test-Time Alignment for Personalization.** Our work builds upon the paradigm of *test-time alignment*, which steers the generation process of a frozen LLM at inference time without updating parameters. The core idea is to guide each token selection step towards user-preferred outcomes. Specifically, given a partially generated sequence $y_{<p}$, the standard approach is to sample the next token $y_p$ from the probability distribution of the base LLM $\pi_{\text{base}}(\cdot|y_{<p})$. To incorporate personalization, we introduce a reward function $r(\cdot; \theta)$ parameterized by $\theta$, which is learned online to capture the user preferences. This reward function assigns

a scalar score to any given sequence, with *higher scores indicating better alignment with the preference of the user*.

At each decoding step $p$, we define a scoring function that combines the base model's likelihood with the learned preference reward. For any candidate token $v$ from the vocabulary $\mathcal{V}$, the score is calculated as:

$$\text{Score}(v|y_{<p}) = \pi_{\text{base}}(v|y_{<p}) + \omega \cdot r([y_{<p}, v]; \theta) \quad (1)$$

where $[y_{<p}, v]$ denotes the new sequence formed by appending token $v$ to the prefix $y_{<p}$, and $\omega$ is a hyperparameter controlling the strength of the personalization. The decoding policy then selects the next token by maximizing this score: $y_p = \arg\max_{v \in \mathcal{V}} \text{Score}(v|y_{<p})$. This framework allows the generation to be dynamically steered towards personalized content by optimizing a local, per-token objective. In the context of test-time personalization under data scarcity, the central challenge lies in efficiently learning the reward function $r(\cdot; \theta)$ from the user's online preference feedback. To address this, we adopt the framework of neural dueling bandits.

**Neural Dueling Bandits.** To learn the reward function $r$ from online preference feedback, we frame the problem within the neural dueling bandits framework (Verma et al., 2024). This setting is designed for learning from pairwise preference feedback (e.g., "response A is better than response B"), which is often more reliable and easier for users to provide than absolute scores.

In this framework, a learner iteratively interacts with a user. In each round, it presents a pair of items (i.e., arms), and the user provides feedback indicating which one they prefer. The user's choice is assumed to be governed by the underlying reward function $r$. This relationship is commonly modeled using the Bradley-Terry-Luce (BTL) model (Hunter, 2004; Luce et al., 1959), which states that the probability of preferring arm $a_1$ over arm $a_2$ is given by: $P(a_1 \succ a_2) = \sigma(f(a_1) - f(a_2))$, where $f$ denotes the unknown reward function and $\sigma(z) = 1/(1 + e^{-z})$ is the sigmoid function. To learn complex user preferences in text generation, we adopt a neural network (NN) $r(\cdot; \theta)$ parameterized by $\theta$ to approximate $f$ (Verma et al., 2024).

## 3. The T−POP Algorithm

In this section, we introduce our T−POP algorithm (Fig. 1, Algo. 1), which addresses the cold-start personalization problem for new users. We begin by discussing the high-level insights behind our approach, followed by a detailed breakdown of its components.

### 3.1. High-Level Overview

The core insight behind T−POP is the synergistic integration of test-time alignment with the principles of online learning

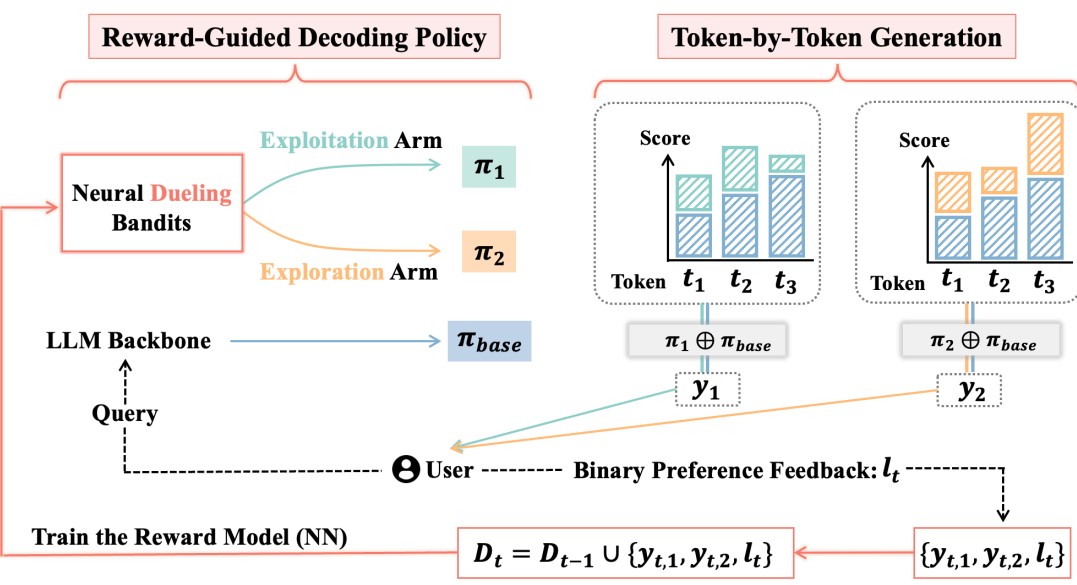

*Figure 1.* An overview of our T-POP for test-time personalization with online preference feedback.

from dueling bandits. Instead of treating personalized text generation and user preference learning as separate phases, T-POP interweaves them into a single, efficient process. The algorithm operates by steering the decoding of a frozen LLM to simultaneously generate two competing sequences in real-time.

This is achieved by applying a dueling bandit policy at *each token-generation step*. The **exploitation sequence** is constructed by greedily following the reward model's current estimate of user preferences (line 11 of Algo. 1). Concurrently, the **exploration sequence** is built by optimistically choosing tokens that balance high estimated reward with high uncertainty (line 12 of Algo. 1). The two completed responses are then presented to the user, who provides feedback on which one they prefer. This feedback is immediately used to update the reward model, improving its alignment with the user preferences. This creates a tight feedback loop: the dueling bandit policy generates **personalized and informative pairs of responses** for learning, and the user feedback immediately refines the reward model, which in turn improves the personalized text-generation policy for the next round of interaction. This entire process requires no fine-tuning of the base LLM, enabling rapid and data-efficient personalization with online feedback.

### 3.2. Online Personalization Loop

T-POP operates over a series of interaction rounds $t = 1, 2, \ldots, T$. The goal in each round is to generate a person-

alized and informative pair of responses $(y_{t,1}, y_{t,2})$, elicit user preference feedback $l_t$, and update the neural network reward model $r(\cdot; \theta)$.

The learning process begins with an initial reward model $r(\cdot; \theta_1)$. In each round $t$, the algorithm generates the pair $(y_{t,1}, y_{t,2})$ based on the current reward model $r(\cdot; \theta_t)$, as detailed in Sec. 3.3. The user then provides a binary preference $l_t = \mathbb{1}_{\{y_{t,1} \succ y_{t,2}\}}$, which is equal to 1 if the response $y_{t,1}$ is preferred over $y_{t,2}$ and 0 otherwise. This new data point is then added to the history $\mathcal{D}_t = \mathcal{D}_{t-1} \cup \{(y_{t,1}, y_{t,2}, l_t)\}$ (line 16 of Algo. 1). Upon receiving this feedback, the parameters of the reward model (i.e., neural network) are updated by minimizing the following loss function over the entire history $\mathcal{D}_t$ (line 17 of Algo. 1):

$$
\begin{aligned}
\mathcal{L}_t(\theta) = - \sum_{(y_1, y_2, l) \in \mathcal{D}_t} & \Big[ l \log \sigma(r(y_1; \theta) - r(y_2; \theta)) \\
& + (1 - l) \log \sigma(r(y_2; \theta) - r(y_1; \theta)) \Big] + \lambda \|\theta\|_2^2,
\end{aligned}
\tag{2}
$$

in which $\sigma(\cdot)$ is the sigmoid function. Of note, minimizing this loss function ((2)) is equivalent to *maximizing the log-likelihood of the preference observations* $\mathcal{D}_t$ *according to the Bradley-Terry-Luce (BTL) model* (Sec. 2), plus a regularization term (Verma et al., 2024). This updated reward model, with parameters $\theta_{t+1} = \arg\min_\theta \mathcal{L}_t(\theta)$, is then used in the next round, enabling continuous improvement of the reward model from user interactions.

---

**Algorithm 1:** `T-POP`

**Input:** Initial reward model parameters $\theta_1$, matrix $V_0 = \lambda I$, number of user interactions $T$, reward weight $\omega$, exploration parameter $\nu$, number of candidate tokens $k$, maximum number of tokens $M$ in a response, observation history $\mathcal{D}_0$.

1 **for** $t = 1, \ldots, T$ **do**
2      Receive the user query $q_t$ in the current round, set $y_{t,1} = [q_t]$, $y_{t,2} = [q_t]$
3      **for** *each token position* $p = 1, \ldots, M$ **do**
4          $\mathcal{V}_p^{(1)} \leftarrow$ top-$k$ tokens conditioned on $y_{t,1}$
5          $\mathcal{V}_p^{(2)} \leftarrow$ top-$k$ tokens conditioned on $y_{t,2}$
6          $\mathcal{V}_p \leftarrow \mathcal{V}_p^{(1)} \cup \mathcal{V}_p^{(2)}$
7          **for** $v \in \mathcal{V}_p$ **do**
8              $score_1(v; \theta_t) \leftarrow \pi_{\text{base}}(v|y_{t,1}) + \omega \cdot r\left([y_{t,1}, v]; \theta_t\right)$
9              $score_2(v; \theta_t) \leftarrow \pi_{\text{base}}(v|y_{t,2}) + \omega \cdot r\left([y_{t,2}, v]; \theta_t\right)$
10          **end**
11          Select token for response 1: $v_{p,1} \leftarrow \arg\max_{v \in \mathcal{V}_p} score_1(v; \theta_t)$
12          Select token for response 2: $v_{p,2} \leftarrow \arg\max_{v \in \mathcal{V}_p} score_2(v; \theta_t) + \omega \cdot \nu \|\nabla r([y_{t,2}, v]; \theta_t) - \nabla r([y_{t,1}, v_{p,1}]; \theta_t)\|_{V_{t-1}^{-1}}$
13          $y_{t,1} \leftarrow [y_{t,1}, v_{p,1}]$, $y_{t,2} \leftarrow [y_{t,2}, v_{p,2}]$
14          $V_{t-1} \leftarrow V_{t-1} + (\nabla r(y_{t,1}; \theta_t) - \nabla r(y_{t,2}; \theta_t))(\nabla r(y_{t,1}; \theta_t) - \nabla r(y_{t,2}; \theta_t))^\top$
15      **end**
16      Obtain binary user preference feedback $l_t = \mathbb{1}_{\{y_{t,1} \succ y_{t,2}\}}$ and update history: $\mathcal{D}_t = \mathcal{D}_{t-1} \cup (y_{t,1}, y_{t,2}, l_t)$;
17      Train NN using history $\mathcal{D}_t = \{(y_{s,1}, y_{s,2}, l_s)\}_{s=1,\ldots,t}$ by minimizing loss function $\mathcal{L}_t(\theta)$ ((2)): $\theta_{t+1} = \arg\min_\theta \mathcal{L}_t(\theta)$
18      Update the covariance matrix: $V_t \leftarrow V_{t-1}$
19 **end**

---

**Deployment via Asynchronous Learning.** Contrary to a rigid "collect-then-deploy" paradigm, `T-POP` is designed for continuous, low-latency deployment throughout the interaction. By decoupling model updates from user interactions, `T-POP` can minimize latency increase:

- **Asynchronous Online Updates:** To eliminate the training latency, we implement an asynchronous update strategy. When a user provides preference feedback at round $t$, the reward model update ($\theta_t \rightarrow \theta_{t+1}$) is triggered in a *background thread*. Crucially, during the model update process, our `T-POP` continues to serve subsequent queries *using the latest reward model* $r(\cdot; \theta_t)$. After the model update concludes, the updated reward model $r(\cdot; \theta_t)$ will then be used to serve subsequent user queries. This ensures that the computational cost of training is completely masked from the user experience.

- **Flexible Deployment Mode:** Once the personalization phase concludes (at any arbitrary interaction $t$), `T-POP` transitions to a definitive inference mode. The learned reward model, $r(\cdot; \theta_t)$, is frozen and utilized solely by the exploitation arm. Generation then proceeds via *token-by-token greedy decoding*, where each token is selected to maximize the score in (1) based on the final reward model. This effectively crystallizes the learned preferences into a standard, low-overhead text generator.

### 3.3. Token-by-Token Arm Generation

A key innovation of `T-POP` is its dynamic, token-by-token construction of the dueling sequences, $y_{t,1}$ and $y_{t,2}$, which is achieved by integrating dueling bandits with reward-guided decoding. The pair of sequences is built over $M$ steps (lines 3–13 of Algo. 1), with the exploitation-exploration policy applied at each step to select the next token for each growing sequence.

**Exploitation Sequence.** The first sequence, $y_{t,1}$, represents pure *exploitation*. It is generated to be the best possible response according to the current reward model $r(\cdot; \theta_t)$. At each token position $p$, the next token $v_{p,1}$ is chosen greedily to maximize the reward-guided scoring function from (1):

$$v_{p,1} = \underset{v \in \mathcal{V}_p}{\arg\max} \left(\pi_{\text{base}}(v|y_{t,1}) + \omega \cdot r([y_{t,1}, v]; \theta_t)\right), \quad (3)$$

where $\mathcal{V}_p$ is a set of candidate tokens formed by the top-$k$ tokens from the base LLM (Algo. 1, lines 4-6). This process iteratively builds a sequence aligned with the current reward model $r(\cdot; \theta_t)$.

**Exploration Sequence.** The second sequence, $y_{t,2}$, simultaneously accounts for exploitation and *exploration*. That is, it aims to not only achieve high reward values to align with the user preference (i.e., exploitation), but also generate informative responses with *large uncertainty* to accelerate the learning of the reward model (i.e., exploration). Specifically, at each token position $p$, it selects the next token $v_{p,2}$ by maximizing the sum of the score and a UCB-style exploration bonus:

$$v_{p,2} = \underset{v \in \mathcal{V}_p}{\arg\max} \Big( \underbrace{\pi_{\text{base}}(v|y_{t,2}) + \omega \cdot r([y_{t,2}, v]; \theta_t)}_{\text{Exploitation}}$$
$$+ \underbrace{\omega \cdot \nu \cdot \text{UncertaintyBonus}(v)}_{\text{Exploration}} \Big). \quad (4)$$

The uncertainty bonus term, denoted as $u_t(v)$, is defined as:

$$u_t(v) = \|\nabla r([y_{t,2}, v]; \theta_t) - \nabla r([y_{t,1}, v_{p,1}]; \theta_t)\|_{V_{t-1}^{-1}}. \quad (5)$$

Our generation strategy is grounded in the theoretically principled Neural Dueling Bandit framework (Verma et al., 2024) and the Tokenized Bandit theory (Shin et al., 2025).

**Theoretical Guarantees for Neural Dueling Bandits.** The matrix $V_{t-1}$ (line 14 of Algo. 1) aggregates the gradient information from all previously selected sequences:

$$V_{t-1} \leftarrow V_{t-1} + \big(\nabla r(y_{t,1}; \theta_t) - \nabla r(y_{t,2}; \theta_t)\big)$$
$$\cdot \big(\nabla r(y_{t,1}; \theta_t) - \nabla r(y_{t,2}; \theta_t)\big)^\top \quad (6)$$

This covariance update allows the uncertainty bonus in (5) to measure the epistemic uncertainty of a candidate sequence $[y_{t,2}, v]$ relative to the exploitation arm $[y_{t,1}, v_{p,1}]$. As established by Verma et al. (2024), maximizing this gradient-based bonus ensures that the system efficiently explores the reward parameter space. Under standard regularity assumptions (e.g., bounded norm in a Reproducing Kernel Hilbert Space), this mechanism achieves a cumulative regret bound of $R_T = \tilde{O}(d_{eff}\sqrt{T})$, where $d_{eff}$ is the effective dimension of the neural tangent kernel matrix. This theoretical result guarantees that our reward model converges to the user's true preference with high probability.

**Theoretical Guarantees for Sequential Decoding.** Extending bandit guarantees to token-by-token generation is non-trivial due to the combinatorial search space. However, our approach is supported by the recent findings of Shin et al. (2025), who proved that linear bandit algorithms applied to token-level decoding achieve sublinear regret $R_T = \tilde{O}(L\sqrt{T})$, provided the utility function satisfies the *Diminishing Distance with More Commons (DDMC)* assumption. Here $L$ denotes the maximum sequence length. Therefore, T-POP effectively operationalizes these theoretical principles: the uncertainty bonus steers generation towards sequences that provide significant novel information (exploration), while the reward score ensures alignment (exploitation), theoretically ensuring both sample efficiency and convergence in the sequential decoding setting.

## 4. Experiments

We conduct comprehensive experiments to empirically validate the effectiveness and data efficiency of our T-POP, particularly its ability to achieve rapid personalization in cold-start scenarios. Some experimental details are deferred to App. A due to space constraints.

### 4.1. Experimental Setting

**Models, Datasets and Personalization Attributes.** We conduct experiments on a diverse set of modern open-source LLMs, including Mistral-7B-Instruct-v0.2 (Jiang et al., 2023), Llama-3.1-8B-Instruct (Grattafiori et al., 2024), and Qwen2-7B-Instruct (Yang et al., 2025). Our evaluation suite is built upon four established benchmarks to ensure a comprehensive assessment. We use (1) **HelpSteer** (Wang et al., 2023) for its multi-faceted instruction-following challenges and two subsets of **UltraFeedback** (Cui et al., 2024): (2) **TruthfulQA** (Lin et al., 2021) and (3) **UltraChat**—to evaluate factuality and conversational ability, respectively. To directly measure alignment with user tastes, we also include the (4) **Personal Preference Eval** (Gao et al., 2024) dataset. To simulate diverse real-world user preferences, we evaluate our method across four distinct preference attributes, inspired by prior work (Zhong et al., 2024; Zhang et al., 2025b): *creative*, *verbose*, *concise*, and *uplifting*.

**Baseline Methods.** We compare our T-POP against a suite of strong baselines representing different personalization paradigms. These include the original, unmodified backbone LLM (**Base**); the backbone guided only by prompt engineering (**Preference Prompting (Pref)**); and a standard decoding algorithm, **Beam Search (BS16)**, with a beam width of 16. We also compare against two state-of-the-art training-free methods: **Linear Alignment (LA)** (Gao et al., 2024), which linearly updates the model's logits to steer generation, and our primary competitor, **AMULET** (Zhang et al., 2025b), which formulates token-level decoding as an online learning problem for test-time alignment.

**Evaluation Metrics.** Given the subjective nature of personalization, we employ a two-pronged evaluation strategy. Our primary quantitative metric is the **Reward Model Score**. We use the widely used **ArmoRM-Llama3-8B-v0.1** (Wang et al., 2024) to score the alignment of generated responses with the target attribute, following the evaluation methodology of Zhang et al. (2025b). To complement this and capture nuances that a single reward model may overlook, we also adopt **GPT-4o** as a Judge (Ouyang et al., 2022). Following the standard protocol (Dubois et al., 2024), we present GPT-4o with the outputs from T-POP and a baseline, and report the win rate.

During the online interaction phase of our T-POP, we use **GPT-4o** to simulate the user and provide pairwise preference feedback based on the target attribute. The evaluation prompts are adapted from the AlpacaEval standard format.

### 4.2. Main Results

An effective personalization method should generate text that is both **strongly** and **consistently** aligned with user preferences. To ensure a comprehensive evaluation, we assess these two aspects separately. First, we utilize the Reward Model Score (Wang et al., 2024) to quantify the **strength** of personalization (Sec. 4.2.1). Second, to measure **consistency**, we report the win rate against the base LLM

*Table 1.* Score comparison across different datasets, attributes and LLMs. The best score is highlighted in **bold**, and the second best score is highlighted in *italics*.

| Model | Dataset | Creative | | | | | | Verbose | | | | | | Concise | | | | | | Uplifting | | | | | |
|---|---|---|---|---|---|---|---|---|---|---|---|---|---|---|---|---|---|---|---|---|---|---|---|---|---|
| | | Base | Pref | BS16 | LA | Amulet | T-POP | Base | Pref | BS16 | LA | Amulet | T-POP | Base | Pref | BS16 | LA | Amulet | T-POP | Base | Pref | BS16 | LA | Amulet | T-POP |
| Mistral-7B | HelpSteer | 0.30 | 0.30 | 0.34 | 0.36 | *0.39* | **0.48** | 0.27 | 0.27 | *0.31* | *0.31* | 0.30 | **0.40** | 0.41 | 0.42 | 0.50 | *0.52* | *0.52* | **0.59** | 0.33 | 0.33 | 0.39 | 0.40 | *0.41* | **0.50** |
| | Personal | 0.34 | 0.34 | 0.35 | 0.38 | *0.42* | **0.47** | 0.30 | 0.30 | 0.30 | 0.30 | 0.30 | **0.39** | 0.47 | 0.49 | 0.50 | *0.54* | 0.53 | **0.65** | 0.41 | 0.42 | 0.42 | 0.45 | *0.46* | **0.52** |
| | Truthful QA | 0.32 | 0.33 | 0.34 | 0.38 | *0.41* | **0.51** | 0.30 | 0.31 | 0.31 | *0.33* | 0.32 | **0.43** | 0.41 | 0.44 | 0.47 | *0.51* | 0.49 | **0.54** | 0.36 | 0.38 | 0.39 | *0.47* | *0.47* | **0.54** |
| | Ultra Chat | 0.34 | 0.35 | 0.35 | 0.36 | *0.38* | **0.47** | 0.31 | 0.31 | 0.31 | 0.32 | 0.31 | **0.39** | 0.45 | 0.46 | 0.47 | 0.49 | *0.51* | **0.61** | 0.38 | 0.39 | 0.39 | 0.41 | *0.42* | **0.50** |
| | Average | 0.32 | 0.33 | 0.34 | 0.37 | *0.40* | **0.48** | 0.30 | 0.30 | 0.31 | *0.32* | 0.31 | **0.40** | 0.43 | 0.45 | 0.48 | *0.52* | 0.51 | **0.60** | 0.37 | 0.38 | 0.40 | 0.43 | *0.44* | **0.51** |
| Qwen2-7B | HelpSteer | 0.34 | 0.34 | 0.35 | 0.35 | *0.36* | **0.50** | 0.31 | 0.32 | *0.33* | *0.33* | 0.30 | **0.44** | 0.43 | 0.48 | 0.50 | 0.57 | *0.59* | **0.60** | 0.38 | 0.38 | 0.39 | 0.39 | *0.41* | **0.52** |
| | Personal | 0.33 | 0.34 | 0.34 | 0.37 | *0.41* | **0.49** | *0.31* | *0.31* | *0.31* | 0.30 | 0.28 | **0.43** | 0.41 | 0.48 | 0.49 | 0.53 | *0.54* | **0.65** | 0.40 | 0.42 | 0.42 | *0.43* | 0.42 | **0.55** |
| | Truthful QA | 0.32 | 0.33 | 0.33 | 0.34 | *0.36* | **0.53** | 0.30 | 0.31 | 0.32 | *0.33* | 0.32 | **0.47** | 0.41 | 0.46 | 0.50 | **0.54** | 0.51 | *0.53* | 0.36 | 0.38 | 0.39 | 0.44 | *0.45* | **0.58** |
| | Ultra Chat | 0.34 | 0.34 | 0.34 | 0.35 | *0.36* | **0.47** | 0.31 | 0.32 | *0.33* | 0.32 | 0.31 | **0.44** | 0.40 | 0.45 | 0.46 | 0.54 | *0.57* | **0.62** | 0.38 | 0.39 | 0.39 | *0.40* | 0.39 | **0.54** |
| | Average | 0.33 | 0.34 | 0.34 | 0.35 | *0.37* | **0.50** | 0.31 | *0.32* | 0.32 | 0.32 | 0.30 | **0.45** | 0.41 | 0.47 | 0.49 | *0.55* | *0.55* | **0.60** | 0.38 | 0.39 | 0.40 | *0.42* | 0.42 | **0.55** |
| Llama-3.1-8B | HelpSteer | 0.33 | 0.34 | 0.36 | 0.44 | *0.50* | **0.51** | 0.30 | 0.31 | 0.33 | 0.36 | *0.41* | **0.51** | 0.40 | 0.43 | 0.45 | 0.53 | *0.57* | **0.62** | 0.36 | 0.37 | 0.39 | 0.45 | *0.50* | **0.53** |
| | Personal | 0.35 | 0.36 | 0.36 | 0.46 | **0.62** | *0.52* | 0.31 | 0.31 | 0.31 | 0.35 | **0.49** | *0.46* | 0.39 | 0.44 | 0.45 | 0.53 | **0.67** | *0.66* | 0.42 | 0.44 | 0.43 | 0.49 | **0.61** | *0.55* |
| | Truthful QA | 0.31 | 0.33 | 0.33 | 0.41 | **0.56** | *0.52* | 0.29 | 0.29 | 0.31 | 0.34 | *0.44* | **0.54** | 0.37 | 0.40 | 0.42 | 0.49 | **0.52** | *0.51* | 0.34 | 0.36 | 0.37 | 0.43 | *0.49* | **0.53** |
| | Ultra Chat | 0.33 | 0.34 | 0.34 | 0.42 | **0.57** | *0.50* | 0.31 | 0.32 | 0.32 | 0.36 | *0.41* | **0.49** | 0.38 | 0.41 | 0.41 | 0.48 | *0.53* | **0.60** | 0.37 | 0.38 | 0.38 | 0.44 | *0.48* | **0.52** |
| | Average | 0.33 | 0.34 | 0.35 | 0.43 | **0.58** | *0.51* | 0.30 | 0.31 | 0.32 | 0.35 | *0.44* | **0.50** | 0.38 | 0.42 | 0.43 | 0.51 | *0.57* | **0.60** | 0.37 | 0.39 | 0.39 | 0.45 | **0.54** | *0.53* |

in pairwise comparisons judged by GPT-4o (Sec. 4.2.2).

### 4.2.1. ARMORM SCORES: ANALYSIS OF THE STRENGTH OF PERSONALIZATION

The main quantitative results, presented in Table 1, benchmark T-POP against strong baselines across a wide range of datasets and attributes. The scores in Table 1 underscore the effectiveness of T-POP in achieving stronger alignment. A detailed model-by-model analysis reveals that our algorithm consistently delivers substantial gains over all baselines, including the strongest baseline, AMULET. The performance uplift is most pronounced on Qwen2-7B, where T-POP demonstrates an average improvement of **28.0%** over the second best method, AMULET, across all four preference attributes. This is closely followed by a **19.9%** average gain over AMULET on the Mistral-7B model. On Llama-3.1-8B, the race is highly competitive, with T-POP and AMULET each securing state-of-the-art scores in two of the four preference dimensions; however, T-POP still maintains a marginal edge with a final average score of **0.535** compared to AMULET's **0.5325**. Aggregating these results, T-POP establishes a robust overall average improvement of **14.7%** against AMULET. This persistent and significant performance improvement across diverse models validates the efficacy of our dueling bandit-based test-time personalization framework, which more efficiently captures the nuances of user preferences than other test-time adaptation methods.

Furthermore, we analyze the impact of the number of user interactions (iterations) on the performance of T-POP. To demonstrate the robustness of its learning efficiency, we present results from two distinct experimental settings: the concise attribute on the Personal dataset and the HelpSteer dataset (Fig. 2). As illustrated across both figures, all three models—Llama-3.1-8B, Mistral-7B, and Qwen2-7B—exhibit a remarkably consistent and efficient learning curve. **The reward scores increase sharply within the first 20 iterations in both scenarios**, indicating that T-POP rapidly captures user preferences with minimal feedback, regardless of the specific task. Following this initial surge, performance gains begin to plateau, with the models reaching their peak alignment between 40 and 60 interactions. Subsequently, the scores remain stable or decrease slightly, which can be attributed to potential overfitting. This consistent trend of rapid initial improvement followed by convergence across diverse datasets further validates the data efficiency and swift personalization capability of T-POP.

### 4.2.2. WIN RATE: ANALYSIS OF THE CONSISTENCY OF PERSONALIZATION

To assess the **consistency** of our personalization method, we employ GPT-4o as a judge to perform pairwise comparisons. For each prompt, GPT-4o evaluates which of two responses—one from our method and one from the base LLM—is better aligned with a given personalization attribute. Table 2 presents the results, where each value represents the *win rate* against the base LLM. This metric measures how consistently an algorithm produces a qualitatively superior and personalized response.

The results show that T-POP achieves personalization with

*Table 2.* Win rate of different algorithms against the base LLM in terms of personalization. All values are reported in percentages (%) The best score is highlighted in **bold**, and the second best score is highlighted in *italics*.

| Model | Dataset | Creative | | | | | Verbose | | | | | Concise | | | | | Uplifting | | | | |
|---|---|---|---|---|---|---|---|---|---|---|---|---|---|---|---|---|---|---|---|---|---|
| | | Pref | BS16 | LA | Amulet | T-POP | Pref | BS16 | LA | Amulet | T-POP | Pref | BS16 | LA | Amulet | T-POP | Pref | BS16 | LA | Amulet | T-POP |
| Mistral-7B | HelpSteer | 95.5 | 94.0 | *98.1* | 90.2 | **99.5** | 79.4 | 76.5 | *91.0* | 79.7 | **93.1** | 87.9 | *89.5* | 87.9 | 75.1 | **92.4** | 86.7 | 85.2 | *95.3* | 92.3 | **97.2** |
| | Personal | 97.1 | 94.3 | *98.5* | 96.6 | **99.1** | 85.4 | 75.4 | **96.5** | 87.8 | *92.5* | *95.4* | 94.0 | 93.8 | 71.3 | **96.7** | 86.4 | 85.6 | *94.2* | 90.2 | **98.2** |
| | Truthful QA | 85.4 | 83.0 | *94.5* | 93.3 | **99.6** | 79.1 | 77.5 | *90.1* | 78.7 | **95.4** | *77.4* | **80.5** | 70.3 | 70.8 | 72.3 | 85.8 | 82.9 | 91.9 | 88.2 | **96.5** |
| | Average | 92.6 | 90.4 | *97.0* | 93.4 | **99.4** | 81.3 | 76.5 | *92.5* | 82.1 | **93.7** | 86.9 | **88.0** | 84.0 | 72.4 | *87.7* | 86.3 | 84.6 | *93.8* | 90.2 | **97.3** |
| Qwen2-7B | HelpSteer | *94.0* | 92.8 | 86.1 | 89.9 | **96.6** | *93.2* | 90.9 | 82.9 | 83.3 | **94.0** | 88.2 | 89.5 | *92.8* | 91.6 | **92.0** | 83.8 | 83.5 | 75.2 | *97.7* | **98.2** |
| | Personal | 95.2 | 96.3 | *98.2* | 96.8 | **99.1** | 93.3 | *96.7* | **100** | 73.7 | 90.3 | 95.8 | 97.6 | **99.1** | 92.0 | *98.7* | 83.5 | 89.9 | **95.8** | 91.1 | *91.3* |
| | Truthful QA | *90.1* | 85.4 | 88.2 | 83.9 | **98.5** | 78.2 | 79.5 | *84.3* | 78.9 | **96.0** | 88.9 | 90.1 | **92.2** | *91.2* | 79.0 | 81.9 | 81.2 | 80.8 | *93.0* | **99.1** |
| | Average | *93.1* | 91.5 | 90.8 | 90.2 | **98.1** | 88.2 | 89.0 | *89.1* | 78.6 | **93.4** | 91.0 | *92.4* | **94.7** | 91.6 | 89.9 | 83.1 | 84.9 | 83.9 | *93.9* | **96.2** |
| Llama-3.1-8B | HelpSteer | 97.4 | 96.2 | 97.4 | *97.6* | **98.6** | 91.7 | 91.4 | **97.6** | 94.7 | **97.6** | 89.0 | 89.3 | **94.3** | 86.3 | *92.3* | 89.4 | 88.8 | **99.0** | 97.5 | *97.6* |
| | Personal | 96.3 | 95.1 | 97.1 | **99.8** | *98.9* | 91.4 | 90.6 | 93.8 | **99.6** | *94.5* | 96.2 | 97.0 | 97.2 | *97.3* | **97.4** | 94.1 | 94.0 | *99.6* | **100** | 94.0 |
| | Truthful QA | 94.1 | 92.3 | 97.2 | **99.5** | *97.3* | 87.3 | 86.7 | **96.5** | 93.2 | *95.4* | 71.9 | *76.9* | 74.7 | **85.5** | 68.8 | 82.7 | 82.6 | **95.3** | 92.8 | *93.5* |
| | Average | 95.9 | 94.5 | 97.2 | **99.0** | *98.3* | 90.1 | 86.2 | **96.0** | 95.8 | *95.8* | 85.7 | *87.7* | 87.7 | **89.7** | 86.1 | 88.7 | 88.5 | **97.8** | 96.8 | 95.0 |

remarkable consistency. Across the 36 experimental settings (3 LLMs × 4 attributes × 3 datasets), our T-POP achieves the highest or second-highest average win rate in 31 cases. Crucially, the win rate for T-POP is almost universally above 90%, averaging **94.2%** across all settings. A win rate over 90% signifies a high degree of confidence that T-POP consistently provides correct alignment and personalization, leading to responses that are qualitatively superior to those from the unguided base model. This robust performance indicates that our T-POP is not only powerful but also highly reliable.

In summary, the ArmoRM scores in Table 1 and the win rates in Table 2 jointly demonstrate that **T-POP achieves strong and consistent personalization**.

## 5. Ablation Study

**The Impact of Reward Weight** $\omega$**.** Fig. 3 illustrates the performance of T-POP across varying $\omega$ values. At $\omega = 0.0$ (base LLM), reward scores are lowest. We observe a sharp improvement at $\omega = 0.1$, with performance peaking at $\omega = 1.0$, indicating that a moderate reward signal is highly effective at steering the generation towards the user preference. However, further increasing the weight to $\omega = 2.0$ and 5.0 leads to a noticeable decline. This suggests that excessive reward weights interfere with the inherent capabilities of the backbone model $\pi_{base}$, causing the decoding strategy to myopically optimize for the reward at the expense of coherence and quality. This phenomenon, known as *reward hacking*, results in superficially aligned but low-quality outputs. Consequently, we identify $\omega \approx 1.0$ as the optimal setting to balance personalization strength and generation quality.

*Table 3.* Reward scores of T-POP for models with different sizes.

| Model | Base Score | T-POP Score |
|---|---|---|
| Qwen2-0.5B-Instruct | 0.27 | **0.29** |
| Qwen2-7B-Instruct | 0.37 | **0.51** |
| Llama-3.2-1B-Instruct | 0.28 | **0.44** |
| Llama-3.1-8B-Instruct | 0.35 | **0.55** |

**Impact of Model Size.** To assess the scalability and model-agnostic properties of T-POP, we evaluate its performance on smaller, resource-efficient LLMs. Specifically, we apply T-POP to Qwen2-0.5B-Instruct and Llama-3.2-1B-Instruct, comparing the ArmoRM scores against those of the base models. The results are presented in Table 3, which confirm that T-POP is able to effectively personalize these smaller models. Notably, T-POP delivers a substantial improvement for the Llama-3.2-1B-Instruct model, increasing its alignment score from 0.28 to 0.44. This finding has significant implications, as it demonstrates that our method can dramatically enhance the capabilities of smaller models, enabling them to achieve a level of personalization typically associated with much larger models. This highlights the potential of T-POP for applications with constrained computational resources, such as on-device deployment.

**Robust to the Noise of Human Preference.** Real-world human feedback is inevitably noisy, making robustness a critical requirement for online personalization. To empirically validate this, we conduct stress tests by artificially injecting noise into the online preference labels. We analyze the robustness of T-POP from two key perspectives: noise intensity and noise injection timing.

- **Impact of Noise Intensity:** We randomly flip preference

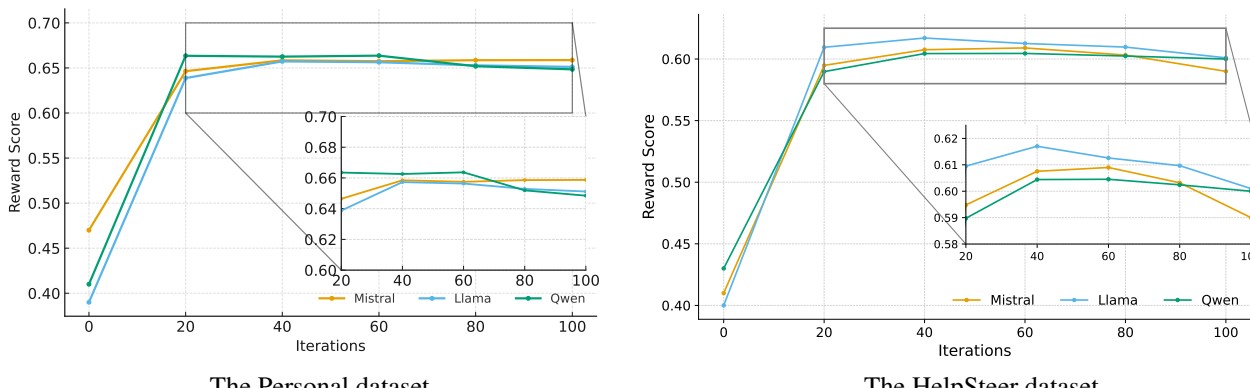

The Personal dataset                                                          The HelpSteer dataset

*Figure 2.* The effect of the number of user interactions on the Reward Score for different models. The results correspond to the concise attribute.

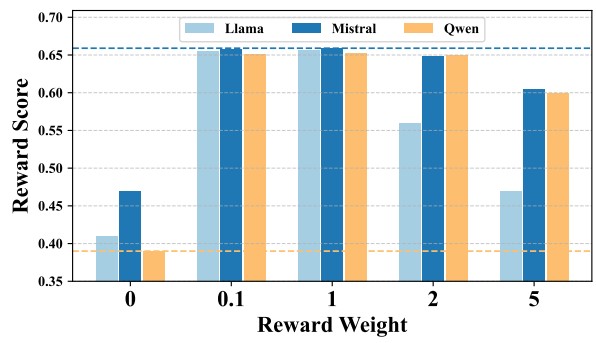

*Figure 3.* The effect of the reward weight ($\omega$) on the alignment performance of `T-POP` across three different backbone models.

labels at rates from $10\%$ to $90\%$ and compare against an untrained baseline. As shown in Table 4, `T-POP` degrades gracefully under moderate noise: with $30\%$ reversed feedback, the ArmoRM score only drops from $0.656$ to $0.649$, remaining far above the untrained RM ($0.451$). This robustness is consistent with recent findings that stable reward-model gradients improve resistance to noisy perturbations (Dong et al., 2025). However, performance declines sharply beyond $50\%$ noise and nearly collapses at $90\%$, showing that `T-POP` tolerates moderate feedback noise but still relies on sufficiently reliable preference labels.

*Table 4.* Impact of Extreme Noise Rates on T-POP Performance.

| Noise Rate | Flipped Samples | ArmoRM Score |
|---|---|---|
| 0.0 (Clean) | 0 | 0.656 |
| 0.1 | 6 | 0.656 |
| 0.2 | 17 | 0.651 |
| 0.3 | 37 | 0.649 |
| 0.5 | 49 | 0.610 |
| 0.7 | 68 | 0.553 |
| 0.9 | 89 | 0.493 |
| Untrained RM | N/A | 0.451 |

- **Impact of Noise Injection Timing:** We further inject $60\%$ local noise into specific learning phases under a comparable global noise ratio. As shown in Table 5, both early and middle-stage injections underperform the uniform baseline, though for distinct reasons. Early-stage noise corrupts the signal before a stable prior forms, yielding a suboptimal trajectory. Middle-stage noise proves slightly more disruptive; introducing contradictory supervision after an initial preference direction is established induces preference oscillation, which destabilizes the learned decision boundary. Conversely, late-stage injection exhibits strong resilience. This indicates that a well-conditioned RM can withstand sporadic corruption once a stable preference landscape is formed. Ultimately, preference consistency in the early-to-middle phases is the primary determinant of alignment quality.

*Table 5.* Impact of Noise Injection Timing (0.2 Global Noise).

| Injection Phase | Global | Local | Flipped | ArmoRM |
|---|---|---|---|---|
| Uniform (Baseline) | 0.2 | – | 17 | 0.651 |
| Early (Iter 1–34) | 0.2 | 0.6 | 21 | 0.645 |
| Middle (Iter 35–67) | 0.2 | 0.6 | 21 | 0.643 |
| Late (Iter 68–100) | 0.2 | 0.6 | 21 | 0.656 |

**Impact of the Uncertainty Bonus.** To validate the effectiveness of our gradient-based uncertainty metric (5), we conducted an ablation study comparing `T-POP` against heuristic exploration strategies, including Entropy Bonus and Boltzmann Exploration. Detailed experimental setups and full numerical results are provided in Appendix C.2. In summary, `T-POP` significantly outperforms these baselines (e.g., achieving a +0.13 gain vs. +0.06 for Boltzmann). This performance gap underscores a critical theoretical distinction: while heuristics like entropy primarily capture *aleatoric uncertainty* (inherent randomness in token prediction), our metric utilizes the gradient norm to quantify *epistemic uncertainty* regarding the user's preference parameters. To efficiently solve the cold-start problem, the

system must explore regions where the *reward model* lacks knowledge, not merely where the *language model* is diverse.

**Impact of the Evaluator Bias.** Relying on a single model (e.g., GPT-4o) for both feedback and evaluation risks introducing evaluator bias. As detailed in Appendix C.3, we conduct a comprehensive cross-model evaluation using distinct LLM judges. T-POP maintains exceptional win rates across diverse, independent evaluators, confirming that it achieves genuine personalization rather than merely overfitting to a specific model's stylistic quirks.

**Adaptation to Complex Domains and Intertwined Preferences.** As detailed in Appendix C.4, we evaluate T-POP in highly realistic settings that demand both multi-domain generalization and the balancing of multi-dimensional preferences. Even under a strictly restricted online training budget, T-POP demonstrates exceptional sample efficiency and domain-agnostic learning. Furthermore, to address intertwined preferences with inherent trade-offs, our lightweight reward model bypasses complex feature extraction. Guided purely by online pairwise comparisons, it efficiently learns the optimal hyperplane to balance competing stylistic constraints, consistently outperforming static baselines in multifaceted scenarios.

**Impact of the Candidate Size.** As detailed in Appendix C.5, varying the candidate size reveals a clear exploration-exploitation trade-off. T-POP achieves peak, robust performance within $k \in [20, 40]$. This empirically validates our default $k = 40$, which optimally balances the exploration of personalized trajectories with the linguistic coherence of the base LLM.

**Alignment-Compute Trade-off.** We provide a detailed analysis of the computational cost in Appendix C.6. While T-POP incurs higher inference latency compared to static baselines due to the nature of test-time optimization, we argue this trade-off is justified by the significant performance gains (14.7% improvement) and the critical capability to address the cold-start problem for new users.

## 6. Related Work

### 6.1. Alignment through Reinforcement Learning from Human Feedback

Reinforcement Learning from Human Feedback (RLHF) (Christiano et al., 2017; Ziegler et al., 2019) is the standard paradigm for aligning LLMs with human preferences. The canonical pipeline (Ouyang et al., 2022) involves three stages: 1) supervised fine-tuning (SFT) on high-quality demonstrations; 2) training a reward model (RM) (Stiennon et al., 2020) on a dataset of human-ranked responses; and 3) fine-tuning the SFT model using an RL algorithm such as PPO (Schulman et al., 2017), with the RM providing the reward signal. The computational expense and instability of

PPO-based RLHF have motivated simpler alternatives. For example, direct Preference Optimization (DPO) (Rafailov et al., 2023) bypasses explicit reward modeling by reframing alignment as a direct policy optimization problem. However, these advancements still produce a single, static policy aligned with a pre-collected, offline dataset, often scaled with techniques like RLAIF (Bai et al., 2022).

### 6.2. Personalized Alignment

Since the universal preference model of conventional RLHF is ill-suited for personalization, a dedicated research area has emerged to adapt LLMs to individual users. One approach involves creating large-scale datasets to model diverse preferences by mapping sociodemographics (PRISM (Kirk et al., 2024)) or constructing user personas from psychological traits (ALIGNX (Li et al., 2025), PAPI (Zhu et al., 2025)). A more data-efficient direction models preferences in a compact, low-dimensional latent space, for instance, by representing them as a linear combination of base reward functions (PReF (Shenfeld et al., 2025), multi-objective alignment (Zhou et al., 2023)) or as latent distributions for few-shot adaptation (VPL (Poddar et al., 2024)). The third direction, most aligned with our work, focuses on lightweight, inference-time adaptation of frozen LLMs. These methods steer the decoding process by manipulating the LLM outputs (PAD (Chen et al., 2024), LA (Gao et al., 2024), decoding-time realignment (Liu et al., 2024)), reframing token generation as an online learning problem (AMULET (Zhang et al., 2025b)), or directly modifying the internal states of the LLMs such as the attention head activations (PAS (Zhu et al., 2025)).

## 7. Conclusion

In this work, we tackled the challenge of cold-start personalization for LLMs, a critical barrier to deploying customized AI assistants for new users. We introduced T-POP, a novel algorithm that enables rapid, real-time personalization by learning directly from online pairwise preference feedback. By synergistically integrating test-time alignment with dueling bandits, T-POP steers the decoding process of a frozen LLM to simultaneously exploit learned preferences and efficiently explore for new ones. Our extensive experiments demonstrate that T-POP achieves significant performance gains over existing baselines with minimal user interaction, confirming its data efficiency and effectiveness for swift personalization. Future work could explore extending this framework to handle more complex feedback structures or adapt to long-term shifts in user preferences.

## Impact Statement

This paper presents work whose goal is to advance the field of Machine Learning. There are many potential societal consequences of our work, none which we feel must be specifically highlighted here.

## Acknowledgements

This work was supported in part by the National Natural Science Foundation of China (Grant Nos. 62506319, 62477012, 62506024), the Guangdong Basic and Applied Basic Research Foundation (Grant No. 2026A1515030032), the Shenzhen Science and Technology Program (Grant No. JCYJ20250604141031003), the Pearl River Talent Program of Guangdong Province (Grant No. 2024QN11X069), the AI for Science Program of the Shanghai Municipal Commission of Economy and Informatization, China (Grant No. 2025-GZL-RGZN-BTBX-01014), the Youth S&T Talent Support Programme of Guangdong Provincial Association for Science and Technology (Grant No. SKXRC2025466), and the General Research Fund (GRF 16209124).

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

## A. Statement on LLM Usage

The authors utilized LLMs solely as writing assistants to improve the grammar, clarity, and readability of this paper. All intellectual contributions, including the core ideas, methodology, and analysis of results, were conducted by the human authors.

## B. More Details on the Experimental Setting

**Reward Model Architecture.** The lightweight reward model, $r(\cdot; \theta)$, is implemented as a simple Multi-Layer Perceptron (MLP) head. This network takes the final hidden-state embeddings from the backbone LLM for a given sequence as input. The MLP consists of one hidden layer with a size of 1024, and all hidden layers utilize the ReLU activation function.

**Diagonal Approximation.** Following the common practice in neural bandits (Zhang et al., 2021; Zhou et al., 2020), we use diagonal approximation to approximate the matrix $V_{t-1}$ ((5)). This allows us to significantly reduce the computational complexity and memory cost of our algorithm.

**Datasets Description.** Since T-POP is a training-free framework, we use the collected data solely for evaluation purposes. Our evaluation suite is constructed from four established benchmarks, from which we only use the question (and discard the responses) to simulate real-world user interactions. The datasets and their sizes are as follows:

- **HelpSteer** (Wang et al., 2023) is a QA dataset aimed at evaluating the model's capability to follow multi-faceted instructions; we utilize its 1,236 testing instances (Zhang et al., 2025b).
- **UltraFeedback** (Cui et al., 2024) is a comprehensive, high-quality AI feedback dataset. From this, we selected two subsets: **Truthful QA** (Lin et al., 2021), using its 811 testing problems to assess factuality, and **UltraChat**, from which we extracted 3,845 problems to evaluate conversational ability (Zhang et al., 2025b).
- **Personal Preference Eval** (Personal) (Gao et al., 2024) is used to evaluate user preference alignment; we utilized the original dataset containing 548 testing instances (Zhang et al., 2025b).

*Table 6.* Hyperparameter settings for T-POP.

| Category | Hyperparameter | Value |
|---|---|---|
| *Dueling Bandit Parameters* | Reward weight ($\omega$) | 1.0 |
| | Exploration parameter ($\nu$) | 0.5 |
| | Regularization parameter ($\lambda$) | 1.0 |
| *Reward Model Online Training* | Optimizer | AdamW |
| | Batch size | 8 |
| | Learning rate | 5e-4 |
| | Epochs per query | 100 |
| | Training Iteration | 100 |
| | Weight decay schedule | $1/(N+50)$ |
| *Decoding Parameters* | Max new tokens | 128 |
| | Candidate tokens ($k$) | 40 |

**Hyperparameters.** The key hyperparameters used for the training of the reward model and the dueling bandit component of T-POP throughout our experiments are listed in Table 6. For the weight decay schedule, $N$ denotes the number of training data points.

**Embedding and Judge Models.** The reward model utilizes embeddings from the Qwen/Qwen3-Embedding-0.6B model (Zhang et al., 2025a). For all experiments requiring preference evaluation, including the simulation of user feedback during the online learning phase and the final win-rate judgments, we employ openai/GPT-4o (OpenAI et al., 2024).

---

**Preference Attribute Descriptions**

PREFERENCE ATTRIBUTES:
"creative": "Your answer should be creative as much as possible.",
"verbose": "Your answer should be verbose as much as possible.",
"concise": "Your answer should be concise as much as possible.",
"uplifting": "Your answer should be uplifting as much as possible."

---

*Figure 4.* Natural language descriptions for the personalized preference attributes.

**Judgement Prompt Template.**    To ensure a consistent and reproducible method for both simulating user feedback and performing the final evaluation, we utilized a structured prompt template adapted from the AlpacaEval format (Dubois et al., 2024). Fig. 4 shows the natural language descriptions for the four core preference attributes used in our experiments. These descriptions serve as the concrete personalization goal.

Fig. 5 displays the main judgment prompt template provided to GPT-4o. In practice, a specific attribute description from Fig. 4 is inserted into the `{attribute description}` field of the main template. The complete prompt then instructs GPT-4o to act as an AI assistant and select which of the two provided responses better embodies the target attribute. This mechanism was used for two critical functions: (1) to generate the online pairwise preference feedback required by `T-POP` during its learning phase, and (2) to conduct the final win-rate evaluations against baseline models, as presented in Section 4.2.

---

**GPT-4o Judgement Prompt Template**

Input:
You are an AI assistant that helps determine which response better aligns with a given attribute preference. Given a specific attribute preference, select the response from assistant A or B that better embodies this attribute.Focus on how well each response aligns with the specified attribute, not general quality. Declare your choice by using the format: "[[A]]" if you believe assistant A's response better aligns with the attribute, or "[[B]]" if assistant B's response better aligns with the attribute.
[Target Attribute]
{attribute}: {attribute description}
[User Question]
{query}
[The Start of Assistant A's Answer]
{response 1}
[The End of Assistant A's Answer]
[The Start of Assistant B's Answer]
{response 2}
[The End of Assistant B's Answer]
[Task] Which response better aligns with the "{attribute}" attribute? Consider how well each response embodies the characteristic described above.

Output:
[[A]] or [[B]]

---

*Figure 5.* The prompt template used to instruct GPT-4o for preference simulation and win rate evaluation.

## C. More Ablation Experiment Results

In this section, we provide additional ablation studies to further validate the efficiency and effectiveness of T-POP. Unless otherwise stated, all experiments in this section are conducted using the **Llama-3.1-8B-Instruct** backbone on the **Personal**

dataset for the **concise** attribute.

## C.1. Analysis of Cold-Start Performance (Early Iterations)

To rigorously evaluate T-POP's capability in addressing the cold-start problem, we analyzed its performance at extremely early stages of user interaction ($T = 5$ and $T = 10$). Table 7 compares the ArmoRM scores of T-POP against baselines.

Remarkably, with only **5 user interactions**, T-POP achieves a reward score of 0.56, which already surpasses the strong training-free baseline Linear Alignment (LA, 0.53) and significantly outperforms static methods like Prompting (0.44). By $T = 10$, the performance gap further widens, demonstrating T-POP's ability to rapidly adapt to user preferences with minimal data.

*Table 7.* Performance comparison at early interaction stages (Proof of Rapid Adaptation).

| Method | Base | Pref | BS16 | LA | Amulet | T-POP (Iter=5) | T-POP (Iter=10) | T-POP (Converged) |
|---|---|---|---|---|---|---|---|---|
| **ArmoRM Score** | 0.39 | 0.44 | 0.45 | 0.53 | **0.67** | 0.56 | 0.63 | *0.66* |

## C.2. Ablation on Exploration Strategies

A core contribution of `T-POP` is the construction of the "Exploration Sequence" ($y_{t,2}$) guided by a principled, gradient-based uncertainty bonus. To rigorously validate this design, we compared `T-POP` against three heuristic exploration strategies that lack explicit knowledge of the reward model's parameter uncertainty.

**Baseline Setup:**

- **Variant A: Entropy Bonus (Aleatoric Uncertainty).** We replaced our gradient-based metric with a token-level entropy term: $\text{Bonus}(v) = -P(v) \log P(v)$. This strategy targets tokens where the *Base LLM* is uncertain (high randomness), rather than where the preference estimate is uncertain.

- **Variant B: Boltzmann Exploration (Noisy Exploitation).** Instead of an explicit information-seeking bonus, we employed High-Temperature Sampling ($T_{high} = 1.5$) on the reward-guided logits. The token selection follows:

$$v_{p,2} \sim \text{Softmax}\left(\frac{\log \pi_{base}(\cdot|y_{t,2}) + \omega \cdot r([y_{t,2}, \cdot]; \theta_t)}{T_{high}}\right)$$

  This induces diversity through randomness but lacks a directed mechanism to reduce epistemic uncertainty.

- **Variant C: Random.** The exploration arm is generated via random sampling from the base LLM, serving as a performance lower bound.

**Analysis of Results.** We performed the experiments using the Llama-3.1-8B-Instruct backbone on the Personal (Gao et al., 2024) dataset ("concise" attribute), with the model trained online for 20 iterations. The results are summarized in Table 8.

As hypothesized, Entropy Bonus yields only a marginal improvement (+0.02) over the random baseline. This empirical result supports our theoretical argument in the main text: high entropy merely indicates that the *LLM* is "confused" between multiple tokens (Aleatoric Uncertainty), which does not necessarily correspond to a sample that provides information about the user's preferences. In contrast, `T-POP`(Ours) achieves a significant gain (+0.13). By utilizing the gradient norm—which serves as a proxy for the Fisher Information Matrix in identifying *Epistemic Uncertainty*—our method actively directs exploration toward regions that maximize information gain for the reward model parameters $\theta$, rather than just generating diverse text.

## C.3. Mitigating Evaluator Bias via Cross-Model Evaluation

A critical concern in online personalization is the risk of "evaluator bias" if the same model (e.g., GPT-4o) is used for both generating online feedback and conducting the final evaluation. To rigorously verify the robustness of our alignment and mitigate this bias, we employ a two-fold evaluation strategy:

*Table 8.* Ablation study on different exploration strategies (Iteration 20).

| Method | Final Score (Iter=20) | Improvement over Random |
|---|---|---|
| T-POP (Random) | 0.51 | - |
| T-POP (Entropy) | 0.53 | +0.02 |
| T-POP (Boltzmann) | 0.57 | +0.06 |
| **T-POP (Ours)** | **0.64** | **+0.13** |

- **Independent Reward Model:** Our primary metric is ArmoRM, an independent Llama-based reward model. Sharing no architectural lineage with GPT-4o, it inherently breaks the feedback-evaluation loop and provides an unbiased assessment.

- **Cross-Model Evaluation:** To further validate performance, we re-evaluate the win rates against the Base model using independent LLM judges with entirely different training paradigms, specifically DeepSeek-V3.2 and Qwen3-235B.

As shown in Table 9, T-POP maintains an exceptional average win rate of 95.5% across all independent evaluators, consistently matching or exceeding the strongest baseline (AMULET). Notably, even under stringent evaluation from differing architectures, performance does not degrade significantly. This multi-model consensus empirically proves that T-POP achieves genuine personalization, rather than merely overfitting to the specific stylistic preferences of GPT-4o.

*Table 9.* Cross-model evaluation of win rates against the Base model.

| Evaluator (Judge) | **T-POP (Ours)** | AMULET | LA | Pref | BS |
|---|---|---|---|---|---|
| GPT-4o (Original) | **97.4%** | 97.3% | 97.2% | 97.0% | 96.2% |
| DeepSeek-V3.2 | **98.2%** | 97.6% | 96.9% | 73.4% | 75.9% |
| Qwen3-235B | 90.9% | **91.2%** | 83.2% | 81.4% | 72.4% |
| **Average** | **95.5%** | 95.4% | 92.4% | 83.9% | 81.5% |

### C.4. Adaptation to Complex Real-World Scenarios

In realistic settings, user interactions span diverse domains, and their specific preferences are rarely single-dimensional. Evaluating isolated attributes or static tasks risks masking biased learning or feature collapse. To rigorously evaluate T-POP under these conditions, we conduct stress tests focusing on both cross-domain generalization and multi-dimensional preference adaptation.

- **Robustness in Diverse Domains and Tasks.** To simulate complex, multi-domain interactions, we construct a Cross-Domain Mixed Dataset by proportionally sampling 500 queries from our four standard benchmarks. To demand rapid preference adaptation, we strictly halve the online training budget to 50 iterations. As shown in Table 10, T-POP demonstrates superior cross-model robustness and domain-agnostic learning. By decoupling the reward model, it successfully isolates stylistic preferences from task semantics, remaining fundamentally robust even during abrupt topic shifts where heuristic baselines often collapse.

*Table 10.* Performance on the Cross-Domain Mixed Dataset under a restricted budget (50 iterations).

| Backbone Model | **T-POP (Ours)** | AMULET |
|---|---|---|
| Llama-3.1-8B | 0.641 | **0.646** |
| Qwen2-7B | **0.612** | 0.569 |
| Mistral-7B | **0.600** | 0.560 |

- **Adaptation to Multi-Dimensional, Intertwined Preferences.** To address concerns regarding oversimplified preference scopes, we introduce new experiments targeting multi-dimensional, intertwined preferences that possess inherent trade-offs: *Academic & Accessible* (rigor + readability), *Structured & Empathetic* (logic + emotion), and *Creative & Concise*

(expressiveness + brevity).

*Table 11.* ArmoRM Scores on Multi-Dimensional Attributes.

| Complex Attribute | AMULET | T-POP (Ours) |
|---|---|---|
| Academic & Accessible | 0.535 | **0.546** |
| Structured & Empathetic | 0.518 | **0.520** |
| Creative & Concise | **0.616** | 0.612 |
| **Average** | 0.556 | **0.559** |

As demonstrated in Table 11, T-POP effectively handles multi-dimensional adaptation, achieving a higher average ArmoRM score (0.559) and outperforming AMULET in two out of three complex scenarios. Since the underlying embedding model already encodes a rich semantic space, our lightweight reward model bypasses the need to learn feature extraction from scratch. Instead, guided purely by online pairwise comparisons, it smoothly learns the optimal hyperplane that balances competing stylistic constraints, confirming its capability for deep, complex adaptation.

### C.5. Impact of Candidate Size

To analyze how the candidate size $k$ impacts the personalization performance, we evaluate the converged ArmoRM scores across different $k$ values. As shown in Table 12, the results exhibit a clear inverted-U shape, reflecting the fundamental exploration-exploitation trade-off during test-time alignment.

*Table 12.* Impact of candidate size ($k$) on T-POP's performance.

| Candidate Size ($k$) | 10 | 20 | 40 | 60 | 80 |
|---|---|---|---|---|---|
| ArmoRM Score | 0.642 | 0.659 | 0.656 | 0.642 | 0.633 |

Specifically, the impact of $k$ can be categorized into three regimes:

- **Under-exploration ($k = 10$):** A restricted candidate set limits the exploration arm's ability to discover optimal personalized trajectories, leading to a suboptimal alignment score (0.642).

- **Over-exploration ($k \geq 60$):** An excessively large $k$ introduces a long tail of low-probability tokens. This degrades linguistic coherence and pushes the generated responses outside the base LLM's semantic boundaries, causing performance to drop significantly (0.633 at $k = 80$).

- **Sweet Spot ($k \in [20, 40]$):** This optimal range provides enough high-quality candidates to explore meaningful personalization while safely remaining within stable semantic boundaries.

Ultimately, the performance peaks and remains highly robust for $k \in [20, 40]$, which empirically validates our default setting of $k = 40$ in the main experiments. **Justification: The Alignment-Compute Trade-off.** As discussed in prior research on test-time alignment (Khanov et al., 2024), an "Alignment-Compute Trade-off" is often unavoidable when moving from static to dynamic systems. While AMULET is faster, it is a *static* baseline that lacks the capability to adapt to new users without retraining. In contrast, T-POP effectively solves the **Cold-Start Problem**, enabling immediate personalization for unseen users. We argue that the marginal increase in computational cost is a justified investment, as it exchanges moderate latency for significant gains in alignment quality (14.7% improvement) and data efficiency.

### C.6. Inference Latency Analysis

In this section, we present a comprehensive evaluation of the computational overhead introduced by T-POP, comparing it against the state-of-the-art baseline, AMULET.

**Latency Breakdown.** As reported in Table 13, T-POP incurs a higher latency compared to AMULET. This overhead is architectural and expected; it primarily stems from two dynamic components: (1) the embedding phase for retrieving user preferences, and (2) the forward pass of the lightweight Reward Model (RM) required for test-time guidance.

Crucially, however, the **token-level latency** remains within the same order of magnitude (0.18s for T-POP vs. 0.09s for AMULET). This indicates that while the initial processing (query-level) takes longer, the generation speed remains viable for real-time interactive systems.

*Table 13.* Wall-Clock Inference Time Comparison.

| Metric | AMULET | T-POP (Ours) |
|---|---|---|
| Query-level Latency | 11.25 s | 23.26 s |
| Token-level Latency | 0.09 s | 0.18 s |

