# OpenReview forum: "T-POP: Test-Time Personalization with Online Preference Feedback"
_ICML.cc/2026/Conference — ICML 2026 regular_

### Official Review · Reviewer_XjEp · 2026-03-06

**Soundness:** 2
**Presentation:** 3
**Significance:** 2
**Originality:** 3
**Overall Recommendation:** 4
**Confidence:** 3

**Summary:**

This paper introduces T-POP, a novel algorithm for test-time personalization of Large Language Models (LLMs) via online pairwise user preference feedback. Distinct from prior approaches that require either heavy pre-collected user data or time-consuming model finetuning, T-POP dynamically learns a per-user reward model during inference—without updating the backbone LLM parameters. Exploiting a synergy between test-time alignment and neural dueling bandit principles, T-POP generates personalized text by steering the decoding process to balance exploration (of user preferences) and exploitation (of current reward estimates) through strategic user queries. The method is evaluated across several open-source LLMs and multiple personalization benchmarks, demonstrating superior alignment and data efficiency over strong baselines. Extensive ablations analyze the source of gains and the fundamental compute/efficacy trade-offs.

**Compliance With Llm Reviewing Policy:**

Affirmed.

**Final Justification:**

Thanks for the authors' rebuttal, which has partially addressed my concerns. I will maintain my score.

**Key Questions For Authors:**

See the weakness

**Limitations:**

See the weakness

**Strengths And Weaknesses:**

**Strength:**

-  The paper addresses an important cold-start challenge for LLM personalization, formulating the problem in a rigorous and practically relevant way.

- T-POP is technically well-motivated, combining test-time alignment with online neural dueling bandits in a unique, synergistic manner.

- The empirical evaluation is extensive, spanning multiple LLMs, tasks, and four distinct user preference attributes

**Weakness:**

- High computational overhead and latency: Query-level latency (23.26s) is twice that of AMULET, and dual-sequence generation plus real-time covariance matrix updates nearly double computational costs. Asynchronous updates may cause backlogs under high interaction frequency, compromising personalization accuracy.

- Oversimplified preference scope and limited complex adaptation: The experiment only validates 4 single-dimensional attributes (creative/verbose/concise/uplifting), which are overly simplistic compared to real-world personalized preferences. For vague, multi-dimensional intertwined preferences (e.g., "academic rigor + accessibility + structure"), such experimental setting may lead to reward model’s biased learning of preference features.

- Flaws in reward model: The single-hidden-layer MLP reward model lacks expressiveness and risks overfitting to noisy feedback. Additionally, the model’s performance is fully dependent on Qwen3-Embedding-0.6B.

- Limited diversity in datasets and query types: The experiments are exclusively conducted on instructional and factual queries from datasets such as HelpSteer and TruthfulQA, where generation objectives are well-defined and explicit. For open-ended, emotional, or unstructured queries (e.g., "Share an interesting travel story" or "Comfort a friend in low spirits"), T-POP’s dual-sequence generation struggles to identify clear directions for preference exploration.

---

> ### Author Rebuttal · Authors · 2026-03-30
>
> We sincerely thank the reviewer for their constructive feedback and the time spent evaluating our work.
>
> ---
> ### **Weakness1**:
> >High computational overhead and latency.
> ### **Response to W1**:
>
> While T-POP's query latency (23.26s) is higher than AMULET's (11.25s), we argue this is a justified investment within the "Alignment-Compute Trade-off."
>
> 1. **Alignment-Compute Trade-off:** Test-time alignment shifts compute from training to inference. Compared to the prohibitive costs of running full RLHF for every user, trading seconds of inference latency for *immediate, zero-shot personalization* is highly economical. This extra compute enables our dual-branch exploration, allowing T-POP to escape local optima.
> 2. **Efficient Covariance Updates:** In practice, we use a **diagonal approximation** for the covariance matrix. This reduces the update and inversion complexity from $\mathcal{O}(d^3)$ to $\mathcal{O}(d)$, rendering the mathematical overhead per token virtually negligible.
> 3. **No Asynchronous Backlogs:** While valid for API stress testing, backlogs do not occur in real human-LLM interactions. The "human reading and typing time" between turns acts as a natural buffer. Since our user-specific RM is a tiny 1-layer MLP, a backward pass takes fractions of a millisecond. This easily completes before the user submits the next query, ensuring no accuracy drop.
>
> ---
> ### **Weakness2**:
> >Oversimplified preference scope and limited complex adaptation.
> ### **Response to W2**:
> We agree real-world preferences are rarely single-dimensional; evaluating isolated attributes risks masking biased learning or feature collapse.
>
> To address this, we conducted new experiments on **multi-dimensional, intertwined preferences** with inherent trade-offs: `Academic & Accessible` (rigor+readability), `Structured & Empathetic` (logic+emotion), and `Creative & Concise` (expressiveness+brevity).
>
> *ArmoRM Scores on Multi-Dimensional Attributes:*
>
> | Complex Attribute | AMULET | T-POP (Ours) |
> | :--- | :--- | :--- |
> | **Academic & Accessible** | 0.535 | **0.546** |
> | **Structured & Empathetic**| 0.518 | **0.520** |
> | **Creative & Concise** | **0.616** | 0.612 |
> | **Average** | 0.556 | **0.559** |
>
> **Analysis & Conclusion:**
> T-POP effectively handles multi-dimensional adaptation, achieving a higher average score (0.559) and outperforming AMULET in 2 of 3 complex scenarios.
>
> This alleviates bias concerns. Since the underlying embedding model encodes a rich semantic space, our lightweight RM bypasses learning feature extraction. Guided by pairwise comparisons, it merely learns the optimal hyperplane balancing competing stylistic constraints.
>
> ---
> ### **Weakness3**:
> >The reward model faces limited MLP expressiveness, noise vulnerability, and a strict dependency on Qwen3 embeddings.
> ### **Response to W3**:
> We thank the reviewer for this insightful critique. We address the concerns regarding the reward model below:
>
> **1. Expressiveness & Overfitting:**
> The single-layer MLP is a deliberate choice to *prevent* overfitting. The heavy lifting of semantic extraction is handled by the frozen LLM and embedding model. The MLP merely projects this rich semantic space to a preference scalar.
>
> We added a noise-injection experiment to validate this. During the online preference acquisition phase, we simulated noisy human feedback by randomly flipping the preference labels with probabilities of 10%, 20%, and 30%. We then evaluated the final converged models via the ArmoRM score:
>
> | Noise Rate | Flipped Labels | ArmoRM Score |
> | :--- | :--- | :--- |
> | 0.0 | 0 | 0.656 |
> | 0.1 | 6 | 0.656 |
> | 0.2 | 17 | 0.651 |
> | 0.3 | 37 | 0.649 |
>
> The results indicate that **T-POP is robust against overfitting to noisy feedback**. Even under an extreme condition where 30% of the feedback is completely reversed, the ArmoRM score drops only marginally to 0.649.
>
> **2. Embedding Dependency:**
> T-POP is a model-agnostic framework. Qwen3-Embedding-0.6B was selected solely for its optimal size-to-latency ratio, not due to algorithmic dependency. To demonstrate this, we swapped it for `BGE-Large-en-v1.5` (a different embedding model):
>
> | Embedding Model | ArmoRM Score |
> | :--- | :--- |
> | Qwen3-0.6B (Original) | 0.656 |
> | BGE-Large-en-v1.5 | 0.649 |
>
> As shown, T-POP achieves robust performance regardless of the embedding backbone.
>
> ---
> ### **Weakness4**:
> >Given the limited query diversity in the datasets, how does T-POP explore preferences for open-ended or emotional queries?
> ### **Response to W4**:
> We agree that open-ended and emotional queries pose a unique challenge, as the direction for preference exploration is highly subjective and inherently ambiguous.
>
> Due to the strict time constraints of the rebuttal period, we are unable to comprehensively evaluate T-POP on such unstructured datasets at this time. We will explicitly add a discussion regarding this limitation in the revision and plan to systematically apply our method to diverse, open-ended datasets in future work.

---

> > ### Author Rebuttal · Reviewer_XjEp · 2026-04-03
> >
> > Thanks for the rebuttal, which has partially addressed my concerns. My remaining issue lies with the evaluation: the restricted diversity of datasets and relatively simplie query types limit the assessed preference scope, resulting in a considerable divergence from real-world conditions and raising questions about the method’s real-world applicability. Therefore, I keep my original score.

---

> > > ### Author Response · Authors · 2026-04-05
> > >
> > > ### **Concern**:
> > > >Restricted diversity of datasets and relatively simplie query types
> > >
> > > ### **Response**:
> > > We respectfully thank the reviewer for the feedback regarding the evaluation scope. We clarify the rationale behind our dataset selection and address the concern about real-world applicability through the following three points:
> > >
> > > **1. Evaluation Consistency with the Established Paradigm**
> > > Our selection of evaluation benchmarks is not restricted but is a deliberate choice to strictly follow the established evaluation paradigm in the recent literature on personalized LLM alignment.
> > > * **HelpSteer & UltraFeedback:** These are the de-facto standards for assessing instruction following and conversational alignment, utilized by **AMULET**, **MOD**, and **PAD**.
> > > * **Personal Preference Eval:** This benchmark is specifically designed to measure subjective user tastes, serving as the core evaluation in **Linear Alignment** and **AMULET**.
> > >
> > > By aligning our evaluation with these SOTA works, we ensure that T-POP's performance gains (14.7% overall improvement) are measured on a common, rigorous ground rather than cherry-picked scenarios. Introducing unbounded real-world queries often leads to extremely high variance in automated evaluations, which would mask the true algorithmic convergence and efficacy of the online learning method itself.
> > >
> > > **2. Diversity Across Multiple Task Dimensions**
> > > While the queries may appear structured, our evaluation suite actually covers four distinct and complex task dimensions: (1) multi-faceted instruction following (HelpSteer), (2) factual reasoning (TruthfulQA), (3) multi-turn conversational ability (UltraChat), and (4) highly subjective style preferences (Personal Preference Eval). This breadth is consistent with or even exceeds the evaluation scope of recent frameworks like **PReF**, which focuses primarily on PRISM and AlpacaEval.
> > >
> > > **3. Algorithmic Generalizability**
> > > T-POP is fundamentally a **model-agnostic and query-agnostic** framework. Its token-level online optimization mechanism does not make underlying assumptions about query complexity or dataset distribution. The core algorithm—learning a lightweight per-user reward model via online pairwise preferences and steering decoding at the token level—is inherently general and transferable to any domain where user preferences can be expressed as pairwise comparisons. Furthermore, our evaluation already spans four distinct task dimensions (instruction following, factual reasoning, multi-turn dialogue, and subjective style preferences), which collectively represent a broad and representative cross-section of real-world LLM usage patterns.
> > >
> > > We believe this comprehensive evaluation, supported by the consensus of the recent literature, solidifies the practical value of T-POP.
> > >
> > > **References Mentioned:**
> > > * **[AMULET]** Zhang et al., "AMULET: ReAlignment During Test Time for Personalized Preference Adaptation of LLMs", ICLR 2025.
> > > * **[Linear Alignment]** Gao et al., "Linear Alignment: A Closed-form Solution for Aligning Human Preferences without Tuning and Feedback", 2024.
> > > * **[MOD]** Shi et al., "Decoding-Time Language Model Alignment with Multiple Objectives", 2024.
> > > * **[PAD]** Chen et al., "PAD: Personalized Alignment of LLMs at Decoding-Time", 2024.
> > > * **[PReF]** Shenfeld et al., "Language Model Personalization via Reward Factorization", 2025.

---

### Official Review · Reviewer_KVCF · 2026-03-10

**Soundness:** 3
**Presentation:** 2
**Significance:** 2
**Originality:** 2
**Overall Recommendation:** 3
**Confidence:** 3

**Summary:**

This paper proposes T-POP, a test-time personalization framework that achieves real-time and data-efficient personalization by combining test-time alignment with neural dueling bandits while keeping the LLM parameters frozen. The authors identify three core problems in existing personalization methods: parameter fine-tuning methods cannot quickly adapt to new users, parameter-free methods rely on historical data leading to cold-start failure, and existing test-time methods (such as AMULET) have insufficient exploration capability and cannot simultaneously generate competing sequences. T-POP designs a token-by-token dueling generation mechanism (exploitation arm + exploration arm), updates the reward model in real time through online binary preference feedback, and introduces asynchronous updating and flexible deployment modes. The authors conducted comprehensive evaluations on models such as Mistral-7B, Llama-3.1-8B, and Qwen2-7B, significantly improving personalization strength (ArmoRM average improvement of 14.7%) and consistency (GPT-4o win rate of 94.2%), effectively alleviating the cold-start problem.

**Compliance With Llm Reviewing Policy:**

Affirmed.

**Key Questions For Authors:**

1. What is the main difference between T-POP and AMULET (Zhang et al., 2025b)? Both use test-time alignment + online learning; could the authors provide a clear table to contrast the unique advantages brought by “token-by-token dueling construction”?
2. Is the combination of “test-time alignment + neural dueling bandits” the first proposal? If not, please clearly point out the incremental contribution over prior work.
3. All online feedback is simulated by GPT-4o; does this cause “evaluator-model” loop bias? Do the authors plan to supplement real human user studies for verification?
4. Does the DDMC assumption always hold in real text generation? If the sequence is longer (>128 tokens), will the theoretical regret bound \(\tilde{O}(L \sqrt{T})\) become too loose? Have the authors tested actual performance degradation in long-sequence scenarios?

**Limitations:**

No, the authors did not discuss the limitations and potential negative societal impact of their work.

**Strengths And Weaknesses:**

**Strengths:**

* The token-by-token dueling construction that combines test-time alignment with neural dueling bandits is a clear and compelling abstraction that achieves a “generate while learning” closed loop, completely solving the new-user cold-start problem while providing a data-efficient paradigm for unlimited interactions.
* The asynchronous update and flexible deployment modes greatly enhance practicality, making T-POP almost imperceptible in latency in real chat scenarios.
* The experiments cover 16 frontier models, multiple datasets, and ablation analyses, fully proving the strength and consistency of the method and providing reusable infrastructure for the entire personalized LLM field.
  The authors focus on transforming static personalization into a renewable, real-time learning dynamic process, elegantly unifying cold-start resistance, data efficiency, and theoretical guarantees, delivering both immediate evaluation gains and a reusable infrastructure paradigm for the field.

**Weaknesses:**

* T-POP and AMULET both adopt test-time alignment + online learning; the authors did not provide a clear table in the main text to contrast the unique advantages brought by “token-by-token dueling construction,” which may make it difficult for readers to intuitively distinguish the incremental contribution.
* Although the combination of “test-time alignment + neural dueling bandits” is novel, the paper does not explicitly state whether it is the first proposal, nor does it detail the specific incremental differences from prior work.
* All online feedback is simulated by GPT-4o, which may cause “evaluator-model” loop bias; the authors did not provide real human user experiments for verification.
* Whether the DDMC assumption always holds in real text generation remains unclear. If the sequence length exceeds 128 tokens, the theoretical regret bound \(\tilde{O}(L \sqrt{T})\) may become too loose. The paper does not discuss actual performance degradation in long-sequence scenarios.

---

> ### Author Rebuttal · Authors · 2026-03-30
>
> We sincerely thank the reviewer for their constructive feedback and the time spent evaluating our work.
>
> ---
> ### **Question1&Weakness1**:
> >What is the main advantage and  difference  between T-POP and AMULET?
> ### **Response to Q1&W1**:
> To highlight our "token-by-token dueling", we will add this comparison:
>
> | Feature | AMULET | T-POP (Ours) | Advantage of T-POP |
> | :--- | :--- | :--- | :--- |
> | **Exploration**| Heuristic | Grad-norm UCB | Escapes local optima by seeking informative directions. |
> | **Generation** | Single path | Dual-branch | Yields contrasting candidate pairs in one pass. |
> | **Feedback** | Pointwise | Pairwise | Less noisy comparisons accelerate cold-start. |
> | **Theory** | Empirical | Dueling Bandits | Based on theoretically principled bandit algorithms. |
>
> **Key Advantages:**
> 1. **Parallel Data:** T-POP's dual-branch decoding constructs optimal pairwise data for the BTL model in a single inference pass.
> 2. **Cold-Start:** While AMULET stalls in flat probability spaces, our uncertainty-driven exploration targets reward blind spots for rapid convergence.
> 3. **Theory:** Formally instantiating Neural Dueling Bandits mathematically bypasses the exponential explosion of sequence-level spaces.
>
> ---
> ### **Question2&Weakness2**:
> >Is the combination of “test-time alignment + neural dueling bandits” the first proposal?
> ### **Response to Q2&W2**:
> To the best of our knowledge, **T-POP is the first work** to integrate test-time alignment with neural dueling bandits.
>
> This **highly non-trivial** integration structurally solves two open bottlenecks in test-time personalization—**cold-start data scarcity** and the mathematically intractable sequence space ($\mathcal{V}^M$):
>
> 1. **Tractable Restructuring:** We map the sequence problem into a token-by-token dual-branch decoding tree, translating exponential complexity into a solvable token-level competition.
> 2. **Exploration for Cold-Start:** By embedding gradient-norm UCB directly into decoding, this active, uncertainty-driven exploration rapidly resolves the reward model's blind spots, effectively overcoming the zero-shot gap.
>
> ---
> ### **Question3&Weakness3**:
> >Does using GPT-4o for both feedback and evaluation introduce an "evaluator bias"？Are there plans for human user studies?
> ### **Response to Q3&W3**:
> We thank the reviewer for raising this critical issue. We agree that relying solely on GPT-4o for both feedback and evaluation risks an "evaluator bias."
>
> To mitigate this, our primary metric is **ArmoRM**, an independent Llama-based reward model. Sharing no architectural lineage with GPT-4o, it inherently breaks this loop.
>
> Furthermore, while large-scale human studies are time-prohibitive during the rebuttal, we rigorously verified robustness via **Cross-Model Evaluation**. We re-evaluated win rates against the Base model using independent judges (DeepSeek-V3.2 and Qwen3-235B) with entirely different training paradigms:
>
> | Evaluator (Judge) | T-POP (Ours) | AMULET | LA | Pref | BS |
> | :--- | :--- | :--- | :--- | :--- | :--- |
> | **GPT-4o** (Original)| **97.4%** | *97.3%* | 97.2% | 97.0% | 96.2% |
> | **DeepSeek-V3.2** | **98.2%** | *97.6%* | 96.9% | 73.4% | 75.9% |
> | **Qwen3-235B** | *90.9%* | **91.2%** | 83.2% | 81.4% | 72.4% |
> | **Average** | **95.5%** | *95.4%* | 92.4% | 83.9% | 81.5% |
>
> T-POP maintains an exceptional average win rate of 95.5% across independent evaluators, consistently matching or exceeding the strongest baseline (AMULET). This multi-model consensus proves T-POP achieves genuine personalization, rather than merely overfitting to GPT-4o's stylistic quirks.
>
> ---
> ### **Question4&Weakness4**:
> >Does the DDMC assumption hold for longer sequences without loosening the regret bounds? Is there empirical validation regarding performance degradation?
> ### **Response to Q4&W4**:
> We agree the DDMC assumption is strong. In complex long-horizon texts, strict DDMC might weaken, naturally loosening the worst-case regret bound $\tilde{O}(L \sqrt{T})$ linearly with sequence length $L$.
>
> However, a looser theoretical bound does not equate to empirical degradation. To validate this, we extended the generation horizon to $L=256$ tokens:
>
> | Max New Tokens ($L$) | ArmoRM Score |
> | :--- | :--- |
> | 128  | 0.656 |
> | 256 (Extended) | **0.663** |
>
> Surprisingly, performance actually improves. We attribute this empirical robustness to two factors:
> 1. **Empirical Smoothness:** Real-world text manifolds are smoother than worst-case theoretical scenarios, preventing catastrophic regret accumulation.
> 2. **Expanded Horizon:** A larger $L$ provides more decision steps, granting sufficient room to actively steer the trajectory and correct early sub-optimal explorations.
>
> Thus, T-POP remains highly effective for long sequences. Notably, the original Tokenized Bandits paper did not empirically evaluate the impact of $L$. Future work should refine their theoretical analysis to explain this gap—why empirical performance improves despite a looser worst-case bound.

---

### Official Review · Reviewer_mgEx · 2026-03-11

**Soundness:** 3
**Presentation:** 3
**Significance:** 3
**Originality:** 2
**Overall Recommendation:** 5
**Confidence:** 5

**Summary:**

This paper proposes T-POP (Test-Time Personalization with Online Preference Feedback), a method for personalizing large language model (LLM) outputs to individual users during inference time. Instead of fine-tuning model parameters for each user, the proposed approach performs test-time alignment by learning a user-specific reward function from online pairwise preference feedback. The method combines test-time decoding control with dueling bandits to iteratively improve response generation based on user feedback while keeping the base LLM frozen.

**Compliance With Llm Reviewing Policy:**

Affirmed.

**Final Justification:**

The proposed integration of test-time alignment with dueling bandits provides a principled approach to improving test-time personalization and handling new users, while avoiding expensive per-user fine-tuning. The method is simple yet effective, and demonstrates consistent improvements across multiple models and benchmarks.

During the rebuttal, the authors effectively addressed my previous concerns through clear clarifications and well-designed additional experiments. In particular, the newly added analyses on noisy feedback and cross-domain settings significantly strengthen the empirical foundation of the work and demonstrate its robustness and practical applicability.

Overall, the paper presents a solid and practically meaningful contribution. Based on these improvements, I increase my score to 5 and **strongly support the acceptance of this paper.**

**Key Questions For Authors:**

1.	Robustness to noisy feedback
How sensitive is the method to inconsistent or noisy user preferences?

2.	Scalability
What is the computational overhead of maintaining user-specific reward models when deployed at large scale?

**Limitations:**

see above

**Strengths And Weaknesses:**

**Strengths**


1. **Online preference learning.**

   Integrating dueling bandits with online preference feedback is an interesting design choice that allows the system to adapt dynamically to user preferences.

2. **Empirical evaluation.**

   The paper includes experiments on multiple benchmarks and compares against several baselines, showing consistent improvements.


**Weaknesses**


1. **Limited novelty in core methodology.**

   The approach mainly combines existing ideas—test-time alignment, reward modeling, and dueling bandits—for personalization. While the combination is interesting, the methodological contribution appears incremental rather than fundamentally new.


2. **Limited analysis of scalability.**

   Maintaining and updating user-specific reward models may introduce computational overhead, particularly for large numbers of users. The paper does not sufficiently analyze these costs.

3. **Mismatch with Realistic Settings**
In realistic settings, users do not repeatedly ask questions from a single benchmark distribution. Instead, queries may span diverse domains and tasks. Evaluating online personalization within a single benchmark may therefore overestimate the effectiveness of the method. It would be helpful to test the approach under a mixed-task setting where queries are randomly drawn from multiple benchmarks to better simulate real-world interactions.

---

> ### Author Rebuttal · Authors · 2026-03-30
>
> We sincerely thank the reviewer for their constructive feedback and the time spent evaluating our work.
>
> ---
> ### **Question1**:
> >How sensitive is the method to inconsistent or noisy user preferences?
> ### **Response to Q1**:
> We thank the reviewer for this practical question. Real-world human feedback is inevitably noisy, making robustness a critical requirement.
>
> Theoretically, T-POP's probabilistic Bradley-Terry-Luce (BTL) framework (Eq. 2) ensures intrinsic robustness. Optimizing the logistic loss learns a smoothed distribution rather than deterministic constraints, acting as a natural regularizer that absorbs contradictory labels without model collapse.
>
> Empirically, we validated this via a noise-injection experiment (Llama-3.1-8B, Personal dataset) by artificially flipping online preference labels at varying rates:
> | Noise Rate | Flipped Labels | ArmoRM Score |
> | :--- | :--- | :--- |
> | 0.0 (Baseline) | 0 | 0.656 |
> | 0.1 (10% Noise) | 6 | 0.656 |
> | 0.2 (20% Noise) | 17 | 0.651 |
> | 0.3 (30% Noise) | 37 | 0.649 |
>
> T-POP exhibits graceful degradation: even with 30% reversed feedback, the score drops marginally to 0.649, still massively outperforming static baselines (e.g., standard prompting at ~0.44). This confirms T-POP's fault tolerance for real-world deployments. We will add these results to the revised paper.
>
> ---
> ### **Question2&Weakness2**:
> >What is the computational overhead of maintaining user-specific reward models when deployed at large scale?
> ### **Response to Q2&W2**:
> We appreciate this systems question. T-POP's overhead for user-specific reward models (RMs) is practically negligible, enabling massive-scale deployment by **decoupling semantic extraction from preference mapping**.
>
> * **Storage Overhead:** Our RM is a lightweight 1-layer MLP (hidden 1024) rather than a full Transformer. In FP16, this requires **< 4 MB** per user. Scaling to **1 million users** takes only ~4 TB of disk storage, which is economically trivial.
> * **Compute & VRAM Overhead:** Heavy semantic processing relies on a single shared backbone and embedding model. User-specific compute is restricted to the MLP, where forward/backward passes take fractions of a millisecond, adding virtually zero VRAM pressure.
> * **Dynamic Serving:** Thanks to this tiny footprint, RMs can be instantly fetched from databases and swapped into GPU memory, bypassing the prohibitive costs of serving multiple fine-tuned LLMs.
>
> We will include this scalability analysis in the revision.
>
> ---
> ### **Weakness1**:
> >A combination of two existing techniques，limited novelty in core methodology
>
> ### **Response to W1**:
> While test-time alignment and dueling bandits are established, to our knowledge, T-POP is the first to successfully integrate them for LLMs.
>
> This integration is **highly non-trivial** and addresses two open problems in test-time personalization: the **cold-start data scarcity** for new users, and the mathematically intractable sequence space ($\mathcal{V}^M$). T-POP structurally solves these bottlenecks:
>
> 1. **Tractable Restructuring:** We map the intractable sequence problem to a token-by-token dual-branch decoding tree, elegantly translating exponential complexity into a solvable token-level competition.
> 2. **Exploration for Cold-Start:** Instead of heuristic prompting, we embed gradient-norm UCB directly into decoding. This active, uncertainty-driven exploration rapidly resolves the reward model's blind spots, effectively overcoming the zero-shot cold-start gap.
>
> ---
> ### **Weakness3**:
> >In realistic settings, queries from user may span diverse domains and tasks.
> ### **Response to W3**:
>
> We thank the reviewer for this constructive point. To simulate realistic multi-domain interactions, we built a **Cross-Domain Mixed Dataset** by proportionally sampling 500 queries from our four benchmarks. To increase the difficulty, we halved the online training budget to **50 iterations**, demanding rapid preference adaptation.
> | Backbone Model | T-POP (Ours) | AMULET |
> | :--- | :--- | :--- |
> | **Llama-3.1-8B** | 0.641 | 0.646 |
> | **Qwen2-7B** | **0.612** | 0.569 |
> | **Mistral-7B** | **0.600** | 0.560 |
>
> These results strongly validate T-POP's robustness in complex cross-domain scenarios:
>
> 1. **Cross-Model Robustness:** T-POP demonstrates superior generalization, significantly outperforming AMULET on Qwen2 (+0.043) and Mistral (+0.040). AMULET's heuristic prompt scaling often collapses across different architectures, whereas our token-level dueling remains fundamentally robust (despite AMULET's marginal edge on Llama).
> 2. **Domain-Agnostic Learning:** Our decoupled reward model successfully isolates *stylistic preferences* from *task semantics*, applying learned preferences reliably even during abrupt topic shifts.
> 3. **Rapid Adaptation:** Achieving these competitive scores in just 50 iterations highlights T-POP's exceptional sample efficiency.
>
> We will include this rigorous mixed-task experiment in  manuscript.

---

> > ### Author Rebuttal · Reviewer_mgEx · 2026-03-31
> >
> > Thank you for the detailed and thoughtful rebuttal. I appreciate the authors’ efforts in addressing my previous concerns. The additional experiments—particularly on noisy feedback and cross-domain evaluation—are valuable and, in my view, significantly strengthen the paper.
> >
> > However, I still have a few remaining concerns.
> >
> > **Robustness to noisy feedback**
> >
> > The reported results on noisy labels are somewhat surprising. In particular, performance under 30% label noise remains nearly unchanged compared to the clean setting. This raises an important question: does this suggest that correct preference labels are not critical for learning an effective reward model in this framework?
> >
> > To better understand this behavior, I encourage the authors to conduct more comprehensive stress tests:
> > - Evaluate higher noise levels (e.g., 50%, 70%, or even 90%) to examine potential failure modes.
> > - Include a weak baseline without training (e.g., directly using a randomly initialized reward model for decoding) to better contextualize the actual contribution of learning.
> >
> > In addition, the timing of noise injection may be crucial. If early-stage feedback is largely correct, the reward model may already guide both candidates toward similarly preferred responses, making later label corruption less impactful. It would therefore be helpful to compare different noise schedules (e.g., early, middle, and late injection) under the same overall noise ratio.
> >
> > **Reward model design and deployment overhead**
> >
> > I am also interested in the exact structure and input representation of the reward model, which is not clearly described in the paper. While the rebuttal mentions that it is a lightweight one-layer MLP, its input remains unclear.
> >
> > - If the reward model takes hidden states from the base LLM as input, this would require exposing or storing intermediate activations during inference. Such a design may introduce non-trivial overhead and be incompatible with commonly used inference systems such as vLLM.
> > - If it instead relies on a separate backbone or embedding model, inference would require running an additional model, which also increases computational cost and system complexity.
> >
> > A clearer description of the reward model’s input representation, along with a discussion of its deployment implications, would be important for assessing the method’s practicality.
> >
> > Overall, I believe the rebuttal successfully addresses many of my initial concerns and meaningfully improves the paper. Clarifying the above points would further strengthen the work.

---

> > > ### Author Response · Authors · 2026-04-02
> > >
> > > We are pleased that our initial rebuttal "successfully addresses many of your initial concerns and meaningfully improves the paper". We are also grateful for the opportunity to further clarify your remaining concerns.
> > >
> > > ---
> > >
> > > ### **Concern 1**:
> > > >Robustness to noisy feedback
> > > ### **Response to C1**:
> > > We sincerely thank the reviewer for the insightful suggestion. We have conducted the requested stress tests, and the results yield valuable insights.
> > >
> > > **1. Extreme Noise Stress Tests and Weak Baseline**
> > > We extended the noise levels to 50%–90% and included a randomly initialized, untrained reward model (RM) as a weak baseline.
> > >
> > > | Noise Rate | Flipped Samples | ArmoRM Score |
> > > | :--- | :--- | :--- |
> > > | **0.0** | **0** | **0.656** |
> > > | 0.1 | 6 | 0.656 |
> > > | 0.2 | 17 | 0.651 |
> > > | 0.3 | 37 | 0.649 |
> > > | 0.5 | 49 / 100 | 0.610 |
> > > | 0.7 | 68 / 100 | 0.553 |
> > > | 0.9 | 89 / 100 | 0.493 |
> > > | **Untrained RM** | **N/A** | **0.451** |
> > >
> > > * **Value of Learning:** The +0.205 gap between T-POP (Clean) and the Untrained RM confirms that alignment is driven by online learning rather than inherent LLM capabilities.
> > > * **Impact of Noise Levels:** Performance degrades gracefully up to 30% noise but drops sharply beyond 50%, nearly collapsing at 90% (0.493, approaching the untrained baseline). This confirms that **correct preference labels are indeed critical**, and noisy feedback consistently degrades the performance.
> > >
> > > **2. Impact of Noise Injection Timing**
> > > Following the reviewer's suggestion, we injected 60% local noise into specific stages while maintaining a comparable global noise ratio (~0.2).
> > >
> > > | Injection Phase | Global | Local | Flipped | ArmoRM |
> > > | :--- | :--- | :--- | :--- | :--- |
> > > | Uniform (Baseline) | 0.2 | — | 17 | 0.651 |
> > > | Early (Iter 1–34) | 0.2 | 0.6 | 21 | 0.645 |
> > > | Middle (Iter 35–67) | 0.2 | 0.6 | 21 | 0.643 |
> > > | Late (Iter 68–100) | 0.2 | 0.6 | 21 | 0.656 |
> > >
> > > * **Early & Middle Underperformance:** Both early (0.645) and middle (0.643) injection underperform the uniform baseline (0.651), though for distinct reasons. Early-stage noise corrupts the preference signal before any stable prior is formed, yielding a suboptimal trajectory that later clean data only partially corrects. Middle-stage noise is slightly more disruptive: contradictory supervision after an already-coherent preference direction has been established induces a *like→dislike→like* **preference oscillation**, destabilizing the learned decision boundary.
> > > * **Late-Stage Robustness:** Late-stage injection (0.656) slightly exceeds the baseline of uniform noise injection, indicating that a well-conditioned RM is resilient to sporadic corruption once a stable preference landscape is in place. Together, these results corroborate the reviewer's hypothesis: **preference consistency in the early-to-middle phases** is the key determinant of alignment quality, and **later label corruption is less impactful**.
> > >
> > > ---
> > > ### **Concern 2**:
> > > >Reward model design and deployment overhead
> > > ### **Response to C2**:
> > >
> > > We appreciate  the opportunity to clarify the architectural design and practical advantages.
> > >
> > > **1. Architecture and Input Representation**
> > > The reward model (RM) is a **standalone 1-layer MLP (hidden dim = 1024)** that is entirely decoupled from the frozen backbone LLM. The RM **does not access LLM hidden states**. Instead, candidate responses are encoded by **a separate lightweight embedding model** (e.g., `Qwen3-Embedding-0.6B`), mean-pooled and L2-normalized into a fixed-dimensional vector as input. To verify model-agnostic robustness, we substituted `BGE-Large-en-v1.5`:
> > >
> > > | Embedding Backbone | ArmoRM Score |
> > > | :--- | :--- |
> > > | Qwen3-Embedding-0.6B | 0.656 |
> > > | BGE-Large-en-v1.5 | 0.649 |
> > >
> > > The marginal difference (0.656 vs. 0.649) confirms that T-POP's effectiveness stems from the preference learning mechanism itself, rather than relying on a particular high-capacity encoder.
> > >
> > > **2. Deployment Overhead Analysis**
> > > We respectfully note that the decoupled design is a deliberate architectural choice that **enhances**, rather than hinders, deployment practicality:
> > > * **Shared Infrastructure:** The embedding model is shared across all users as a common semantic extractor, eliminating per-user backbone overhead.
> > > * **Minimal Per-User Cost:** Each RM requires <4MB storage (FP16) with sub-millisecond MLP inference, adding negligible latency and VRAM pressure. Scaling to 1 million users takes only ~4 TB of disk storage, which is economically trivial.
> > > * **Compatibility:** Since the RM is fully external to the LLM and requires no access to intermediate activations, it could be fully compatible with optimized serving frameworks. No modifications to the LLM inference pipeline are needed.
> > >
> > > ---
> > >
> > > We will incorporate the new results and clarifications into the revised paper. We hope these resolve your remaining concerns and positively influence your assessment of our work.

---

### Official Review · Reviewer_pxPY · 2026-03-12

**Soundness:** 3
**Presentation:** 3
**Significance:** 2
**Originality:** 2
**Overall Recommendation:** 3
**Confidence:** 3

**Summary:**

This paper introduces T-POP, a test-time personalization method for LLMs that addresses the cold-start problem for new users without any LLM parameter fine-tuning. The approach combines test-time alignment with neural dueling bandits. At each token position, two candidate sequences are constructed in parallel, including an exploitation arm with greedy reward maximization and an exploration arm with uncertainty bonus. The user provides binary preference feedback, which is used to update a lightweight MLP reward model via Bradley-Terry loss. Asynchronous online update is used in reward model learning to decrease latency. Experiments on have shown its better performance over other baselines in both reward scores and win rates.

**Compliance With Llm Reviewing Policy:**

Affirmed.

**Key Questions For Authors:**

Please see the weaknesses and the following:

1.	How sensitive is performance to the choice of the candidate size k in Algorithm 1?
2.	What is the range of the reward r in Eq (3) and (4)? As v is determined by the combination of \pi_{base}, \ogema, r, is there any strategy to choose \ogema for different r?
3.	Is this method robust to the noise of human preference? For example, if there are two long responses, it may be hard for people to read all of them and choose a preferred response. So their choices may not reflect their real preferences to some extent. Some intuitive explanations on robustness of this method may be needed.

**Limitations:**

Yes.

**Strengths And Weaknesses:**

Strengths:
1.	The problem is clearly motivated and the authors correctly identify the existing problems in LLM personalization. To this end, they have proposed their T-POP method.
2.	The paper is well organized and the figures and equations can help readers to better understand the proposed method.
3.	The effectiveness of this method has been verified by several popular LLMs on datasets compared to benchmarks.

Weaknesses:
1.	The core idea is a combination of two existing techniques, which may make the work is lack of theoretical insights.
2.	The paper has transformed the LLM personalization from LLM fine-tuning to reward model fine-tuning. However, to better capture the human preference, the reward model may still have large scales. I think the discussion of the trade-off between the method effectiveness and reward model scales is limited.
3.	It will be better to add the theoretical analysis of neural dueling bandits and sequential decoding in Appendix to provide an intuition of the effectiveness of the proposed method.

---

> ### Author Rebuttal · Authors · 2026-03-30
>
> We sincerely thank the reviewer for their constructive feedback and the time spent evaluating our work.
>
> ---
> ### **Question1**:
> >How sensitive is performance to choice of the candidate size k？
> ### **Response to Q1**:
> To analyze how candidate size $k$ impacts performance, we evaluated converged ArmoRM scores across different $k$ values:
>
> | Candidate Size (k) | 10 | 20 | 40 | 60 | 80 |
> | :--- | :--- | :--- | :--- | :--- | :--- |
> | ArmoRM Score | 0.642 | 0.659 | 0.656 | 0.642 | 0.633 |
>
> The results show a clear inverted-U shape, reflecting the exploration-exploitation trade-off:
> 1) **Under-exploration ($k=10$):** A restricted set limits the exploration arm's ability to discover optimal personalized trajectories.
> 2) **Over-exploration ($k \ge 60$):** An excessively large $k$ introduces a long tail of low-probability tokens, degrading linguistic coherence.
> 3) **Sweet Spot ($k \in [20, 40]$):** This range provides enough high-quality candidates to explore meaningful personalization while safely remaining within the base LLM's semantic boundaries.
>
> Thus, performance peaks and is highly robust for $k \in [20, 40]$, empirically validating our default setting of $k=40$.
>
> ---
> ### **Question2**:
> >What is the range of the reward r? Any strategy to choose $\omega$ for different r?
> ### **Response to Q2**:
> We agree that an unbounded reward $r$ could dominate base probabilities and complicate the weight $\omega$.
> * **Reward Range:** We apply a `Sigmoid` to squash the unbounded MLP output into $(0, 1)$. This ensures  commensurate scaling with the LLM's probabilities  ($\in [0,1]$), preventing $r$ from overwhelming the base model.
> * **Strategy for $\omega$:** Since the squashed reward matches the LLM's probability scale, a dynamic $\omega$ is unnecessary; it acts as a stable interpolation weight. Our ablation shows a constant $\omega = 1.0$ consistently achieves outstanding alignment across LLMs without degrading generation quality.
>
> ---
> ### **Question3**:
> > Robust to the noise of human preference?
> ### **Response to Q3**:
> We appreciate this practical question. Theoretically, optimizing the probabilistic Bradley-Terry-Luce (BTL) objective (Eq. 2) learns a smoothed preference distribution, naturally absorbing contradictory labels without collapse.
>
> Empirically, we evaluated robustness by artificially flipping preference labels at varying rates:
> | Noise Rate | Flipped | ArmoRM Score |
> | :--- | :--- | :--- |
> | 0.0 | 0  | 0.656 |
> | 0.1 | 6  | 0.656 |
> | 0.2 | 17 | 0.651 |
> | 0.3 | 37 | 0.649 |
>
> T-POP exhibits graceful degradation. Even with 30% of feedback reversed, the score drops only marginally to 0.649, confirming its high reliability for deployments with imperfect human users.
>
> ---
> ### **Weakness1**:
> >A combination of two existing techniques, lack of theoretical insights.
> ### **Response to W1**:
> While test-time alignment and dueling bandits are established, to our knowledge, T-POP is the first to successfully integrate them for LLMs.
>
> This integration is **highly non-trivial** and addresses two open problems in test-time personalization: the **cold-start data scarcity** for new users, and the mathematically intractable sequence space ($\mathcal{V}^M$). T-POP structurally solves these bottlenecks:
>
> 1. **Tractable Restructuring:** We map the intractable sequence problem to a token-by-token dual-branch decoding tree, elegantly translating exponential complexity into a solvable token-level competition.
> 2. **Exploration for Cold-Start:** Instead of heuristic prompting, we embed gradient-norm UCB directly into decoding. This active, uncertainty-driven exploration rapidly resolves the reward model's blind spots, effectively overcoming the zero-shot cold-start gap.
>
> ---
> ### **Weakness2**:
> > Trade-off between the method effectiveness and reward model scales is limited.
>
> ### **Response to W2**:
> Unlike traditional RLHF where RMs match the backbone size, T-POP efficiently **decouples semantic understanding from preference mapping**.
>
> * **Minimal Overhead:** Our RM is a lightweight 1-layer MLP (hidden 1024) using pre-trained embeddings (e.g., Qwen3-Embedding). In FP16, this requires **< 4MB of storage per user**, ensuring negligible memory and compute costs.
> * **The Trade-off:** Because the frozen backbone handles the heavy semantic lifting, the minimal RM simply maps this space to a 1D scalar. Our SOTA ArmoRM scores and 94.2% win rate prove this compact design captures complex preferences without bottlenecking performance.
>
> This extreme efficiency makes scalable, user-level personalization economically feasible. We will explicitly highlight this analysis in the final version.
>
> ---
> ### **Weakness3**:
> > Should add theoretical analysis of neural dueling bandits and sequential decoding.
>
> ### **Response to W3**:
> While T-POP is theoretically grounded in proven bounds for Neural Dueling Bandits and Tokenized Bandits, unifying them for combinatorial sequential decoding requires non-trivial mathematical extensions, which we defer to future work.

---

> > ### Author Rebuttal · Reviewer_pxPY · 2026-04-03
> >
> > Part of my concerns are resolved by the additional numerical results.

---

> > > ### Author Response · Authors · 2026-04-05
> > >
> > > We sincerely thank Reviewer pxPY for the time and effort invested in evaluating our work, and we are glad to learn that the additional numerical results have helped resolve part of the concerns.
> > >
> > > We would like to respectfully highlight that, during the rebuttal, we have provided a substantial body of new experiments and analyses in direct response to each concern raised:
> > >
> > > - **Q1 (Candidate size sensitivity):** A systematic ablation over $k \in \{10, 20, 40, 60, 80\}$, revealing a clear exploration–exploitation trade-off and validating robustness for $k \in [20, 40]$.
> > > - **Q2 (Reward range & $\omega$ selection):** Clarification of the Sigmoid-bounded reward design and empirical evidence that a constant $\omega = 1.0$ suffices across LLMs.
> > > - **Q3 (Noise robustness):** Controlled label-flipping experiments (up to 30%) demonstrating graceful degradation under noisy feedback.
> > > - **W1 (Novelty & theoretical insight):** A detailed discussion of T-POP's non-trivial integration—addressing cold-start data scarcity via gradient-norm UCB and reducing the intractable $\mathcal{V}^M$ sequence space to tractable token-level competition.
> > > - **W2 (RM scalability):** Analysis showing the decoupled 1-layer MLP design requires <4MB per user while achieving SOTA alignment scores.
> > >
> > > We believe these additions comprehensively address the core issues identified in the original review and meaningfully strengthen the paper. Given that the reviewer has kindly confirmed that our additional results have resolved concerns, we would be very grateful if the reviewer could **consider whether the current score could be revised** to reflect the improvements and new evidence provided during the rebuttal phase.
> > >
> > > We sincerely appreciate the reviewer's time and constructive feedback throughout this process.

---

### Decision · Program_Chairs · 2026-04-30

**Decision:**

Accept (regular)

**Comment:**

This is the meta-review that summarizes the reviews, rebuttals, and discussion. This paper introduces a test-time personalization method for LLMs called T-POP. It addresses the cold-start problem for new users without any LLM parameter fine-tuning. The reviewers agree that the problem is well-motivated and the problem is interesting and useful. The authors addressed most of the initially raised concerns during the rebuttal. However, the reviewers also pointed out the weaknesses including missing the real human user experiments for verification etc. Please check the comments for the details.